# Opponent Shaping in LLM Agents

**Marta Emili Garcia Segura[1, 2], Stephen Hailes[1], Mirco Musolesi[1, 2, 3]**
[1]Department of Computer Science, University College London
[2]Centre for Artificial Intelligence, University College London
[3]Department of Computer Science, University of Bologna
{marta.segura.22, s.hailes, m.musolesi}@ucl.ac.uk

## Abstract

Large Language Models (LLMs) are increasingly being deployed as autonomous agents in real-world environments. As these deployments scale, multi-agent interactions become inevitable, making it essential to understand strategic behavior in such systems. A central open question is whether LLM agents, like reinforcement learning agents, can shape the learning dynamics and influence the behavior of others through interaction alone. In this paper, we present the first investigation of opponent shaping (OS) with LLM-based agents. Existing OS algorithms cannot be directly applied to LLMs, as they require higher-order derivatives, face scalability constraints, or depend on architectural components that are absent in transformers. To address this gap, we introduce ShapeLLM, an adaptation of model-free OS methods tailored for transformer-based agents. Using ShapeLLM, we examine whether LLM agents can influence co-players' learning dynamics across diverse game-theoretic environments. We demonstrate that LLM agents can successfully guide opponents toward exploitable equilibria in competitive games (Iterated Prisoner's Dilemma, Matching Pennies, and Chicken) and promote coordination and improve collective welfare in cooperative games (Iterated Stag Hunt and a cooperative version of the Prisoner's Dilemma). Our findings show that LLM agents can both shape and be shaped through interaction, establishing opponent shaping as a key dimension of multi-agent LLM research.

## 1 Introduction

Large language models (LLMs) have evolved rapidly in recent years, demonstrating remarkable capabilities in reasoning, planning and goal-directed behavior that make them increasingly suitable for deployment as autonomous agents (Zhao et al., 2023; Anthropic, 2025a; OpenAI, 2025b; Xi et al., 2025). Already, LLM-based agents are being adopted for complex tasks such as web navigation and code generation (Anthropic, 2025b; OpenAI, 2025a). As deployment scales, these agents will be less likely to operate in isolation. Instead, they will increasingly interact with other learning agents in shared environments, collaborating on tasks, competing for resources, or pursuing independent objectives. There is growing interest in understanding the opportunities and challenges associated with multi-agent LLM systems (Fourney et al., 2024; Ghafarollahi & Buehler, 2025; Pan et al., 2025; Rosser & Foerster, 2025). However, most approaches treat LLMs as static entities, overlooking the strategic dynamics that emerge when agents continuously adapt to one another.

Multi-agent reinforcement learning (MARL) has long been concerned with the interaction of multiple learners in shared environments (Busoniu et al., 2008). A core difficulty in MARL is that agents often treat each other as static parts of the environment, which can yield poor collective outcomes. For instance, in the Iterated Prisoner's Dilemma (IPD, Axelrod & Hamilton (1981)), independent learners reliably converge to mutual defection, which is the worst collective outcome (Harper et al., 2017; Foerster et al., 2018). To mitigate such failures, the field of opponent shaping develops agents that actively anticipate and influence their co-players' learning dynamics, steering learned behavior toward more favorable equilibria (Foerster et al., 2018; Lu et al., 2022). While these methods have proven effective for multiple architectures (e.g., tabular policies and recurrent neural networks), it remains unclear whether they extend to LLM agents. These agents process rich semantic information, exhibit complex reasoning capabilities (Xu et al., 2025), and can adapt their behavior through

in-context learning (Brown et al., 2020; Bubeck et al., 2023). This raises uncertainty about whether traditional shaping methods will transfer to LLM agents.

Our work addresses this gap, being the first exploration of opponent shaping with LLM agents. Understanding the extent to which LLMs can engage in opponent shaping is critical as they are increasingly deployed in real-world, multi-agent settings. This capability has dual implications: LLM agents may be vulnerable to exploitation by adversaries who strategically influence their learning dynamics, while shaping could also be a tool to foster prosocial behavior and enable coordination.

We study whether LLM agents can strategically influence each other's learning dynamics in repeated matrix games, which capture core strategic incentives while allowing precise outcome quantification. As a baseline, we train two LLM agents independently using Proximal Policy Optimization (PPO, Schulman et al. (2017)) to maximize their individual returns. We then introduce opponent shaping by turning one of those agents into a *shaper* that aims to alter the learning dynamics of its co-player. The shaper is trained using *ShapeLLM*, our proposed algorithm that adapts model-free approaches (Lu et al., 2022; Khan et al., 2024) to transformer architectures. We evaluate the efficacy of shaping in both exploitative scenarios, where an agent seeks unilateral advantage, and prosocial scenarios, where shaping fosters cooperation. Our contributions are the following:

- We provide the first investigation of opponent shaping in LLM agents, demonstrating that they can strategically influence each other's learning dynamics through interaction alone.

- We propose *ShapeLLM*, a model-free opponent shaping algorithm for transformer architectures leveraging structured natural language prompts.

- We evaluate shaping across diverse game-theoretic environments and show that LLM agents can successfully exploit opponents in competitive settings and guide interactions toward mutually beneficial outcomes in cooperative ones.

## 2 BACKGROUND

### 2.1 LLM AGENTS

An *LLM agent* can be thought of as any system leveraging LLMs as the core computational unit for reasoning, planning, and decision-making (Sumers et al., 2023). The architectures of these agents vary significantly in the literature, with some systems integrating reasoning frameworks (Yao et al., 2023), memory banks (Vezhnevets et al., 2023), or tool-access via APIs (Schick et al., 2023; Patil et al., 2024), both in single and multi-agent settings (Park et al., 2023; Wang et al., 2024). In the latter, game-theoretic environments provide a natural testbed for studying strategic dynamics. Recent work has used these environments to investigate LLM agents' cooperation (Piatti et al., 2024; Akata et al., 2025), rationality (Fan et al., 2024), and strategic reasoning (Gandhi et al., 2023; Duan et al., 2024; Huang et al., 2025). Beyond observational studies, these environments can be used to train LLMs towards specific objectives such as moral alignment (Tennant et al., 2025). In this work, we employ game-theoretic environments as controlled testbeds with quantifiable outcomes and clear incentive structures to study whether LLM agents can both shape and be shaped by opponents.

### 2.2 FINE-TUNING LLMS WITH REINFORCEMENT LEARNING

The use of reinforcement learning (RL) in LLM training was first popularized through Reinforcement Learning from Human Feedback (RLHF, Ziegler et al. (2019); Stiennon et al. (2020); Ouyang et al. (2022)), which typically employs Proximal Policy Optimization (PPO, Schulman et al. (2017)) as the RL algorithm. PPO is an on-policy, actor-critic method that uses a learned value function for advantage estimation. When applied to LLMs, it is customary to include a Kullback-Leibler (KL) penalty in the reward signal to prevent the model's output distribution from diverging too far from the pre-trained one. Several alternatives to RLHF with PPO have been proposed, including Group Relative Policy Optimization (GRPO, Shao et al. (2024)), which estimates advantages via Monte Carlo rollouts, and Direct Preference Optimization (DPO, Rafailov et al. (2024)), which converts the RLHF objective into a supervised learning loss. All three approaches are typically used in contextual bandit settings (Llama Team, AI@Meta, 2024a; Qwen Team, 2025; Mistral-AI, 2025), where each model response is treated as an independent episode with immediate reward feedback.

The application of multi-turn RL to LLMs remains an active area of research due to challenges in preference collection, reward modeling, and ambiguity in action space definition (Shani et al., 2024; Zhou et al., 2024; Zeng et al., 2025). Nevertheless, underlying algorithms such as PPO are inherently designed to handle temporally structured environments.

## 2.3 OPPONENT SHAPING

The opponent shaping literature is characterized by two primary approaches: methods that explicitly account for the opponent's updates in the agent's learning rule (Foerster et al., 2018; Letcher et al., 2019; Willi et al., 2022), and meta-learning approaches that learn to shape opponents by observing how their actions influence their opponent's parameter updates (Lu et al., 2022; Balaguer et al., 2022; Khan et al., 2024). The most notable method in the first category is Learning with Opponent-Learning Awareness (LOLA, Foerster et al. (2018)). In LOLA, the opponent's learning rule is incorporated into the agent's update, accounting for the effect of the agent's action on the opponent's parameter updates. This method has demonstrated notable successes, such as the emergence of Tit-for-Tat (TFT) in the iterated prisoner's dilemma (IPD) through self-play. However, LOLA faces significant limitations: it assumes knowledge of the opponent's learning rule, relies on high-variance higher order derivatives, and only considers immediate effects on the opponent's updates. While several LOLA refinements have been proposed (Letcher et al., 2019; Willi et al., 2022), they exhibit the same limitations. A distinct line of work proposes an opponent shaping algorithm based on a modified advantage function which includes the opponent's advantage (Duque et al., 2025). While being higher-order-derivative free, this approach requires access to the opponent's value function, distinguishing it from fully model-free methods.

The second category of methods, exemplified by Model-Free Opponent Shaping (M-FOS, Lu et al. (2022)), was developed specifically to overcome these challenges. M-FOS bypasses these limitations, particularly LOLA's myopic perspective, by framing opponent shaping as a meta-learning problem. In doing so, it decouples the task of interacting with the environment from that of influencing the opponent's learning dynamics. This decoupling is achieved via a bi-level agent architecture: an inner agent that interacts with its co-players, and an outer agent that updates or conditions the inner agent's policy. The outer agent operates in a meta-game, where the meta-state consists of all players' parameters, and the meta-action determines the inner agent's policy. Between episodes, other players update their parameters using their respective learning algorithms. This formulation enables the meta-agent to optimize for long-term opponent shaping effects.

While M-FOS has demonstrated strong empirical results, including outperforming LOLA-based agents in the IPD, it presents scalability challenges due to its dual-agent architecture. To address these, Khan et al. (2024) propose SHAPER, which simplifies M-FOS' architecture by collapsing the shaping agent into a single recurrent neural network (RNN). The key insight is the distinction between history and context within opponent shaping. History captures intra-episode information necessary for implementing conditional strategies such as TFT, while context captures inter-episode information about the opponent's learning dynamics. SHAPER captures history through the RNN's inputs and context through its hidden state, which persists across episodes within a trial. This unified architecture eliminates the dual action spaces of M-FOS, allowing the agent to operate directly in the environment's original action space. However, SHAPER is inherently tied to RNN architectures, with its mechanism relying on distinct memory streams for capturing history and context. Recently, Meulemans et al. (2024) proposed COALA-PG, which also employs RNN-based sequence policies in the same meta-learning setting but uses policy gradient algorithms instead of evolution strategies.

## 3 METHODOLOGY

### 3.1 PRELIMINARIES

Agents interact with the environment by generating text. Let $\mathcal{V}$ denote the model's vocabulary, and $w_{1:L} := (w_1, \ldots, w_L)$, with $w_l \in \mathcal{V}, \forall l \in \{1, \ldots, L\}$. At each interaction, the agent's action is a sequence of tokens sampled from:

$$\rho_\theta(w_{1:L} \mid c) = \prod_{l=1}^{L} \rho_\theta(w_l \mid c, w_{<l}), \tag{1}$$

where $w_{<l} := w_{1:l-1}$, the context $c$ is the environment's description, and $L$ is the generation length. For simplicity, we set $L = 1$ and define the distribution $\rho_\theta(w \mid c)$, with $w \in \mathcal{V}$, as our agent's policy. We deliberately avoid constrained decoding (Beurer-Kellner et al., 2024; Ugare et al., 2024) or format-specific fine-tuning, relying instead on textual instructions to guide the model toward the desired format.

We formalize our environments as *repeated normal-form games* (Fudenberg & Tirole, 1991). Let $\mathcal{M} = (I, \{\mathcal{A}_i\}_{i \in I}, \{R_i\}_{i \in I})$ denote a base game, where $I = \{1, \ldots n\}$ represents the set of players. Each player $i \in I$ has an associated action space $\mathcal{A}_i$ and reward function $R_i : \mathcal{A}_1 \times \ldots \times \mathcal{A}_n \to \mathbb{R}$. A repeated normal-form game consists of playing $\mathcal{M}$ for $T$ rounds, where $T$ can be finite or infinite. In this paper, we focus on the finite case. In each round $t$, players simultaneously choose actions $a_i^t$ for $i \in I$. The resulting joint action $\mathbf{a}^t = (a_1^t, \ldots, a_n^t)$ determines the reward $r_i^t = R_i(\mathbf{a}^t)$ that each player receives. The actions $a_i^t$ are sampled from player-specific policies $\rho_{\theta_i}(w \mid f(h^t))$, where $f(h^t)$ is any function of the joint action history $h^t = (\mathbf{a}^1, \mathbf{a}^2, \ldots, \mathbf{a}^{t-1})$ (e.g., the previous joint action $\mathbf{a}^{t-1}$).

## 3.2 Opponent Shaping in LLM Agents

We introduce *ShapeLLM*, a model-free opponent shaping algorithm designed to leverage the natural language capabilities of LLMs. ShapeLLM condenses both history and context into structured natural language prompts, explicitly capturing the two forms of memory required for shaping into one information stream. As in existing model-free algorithms (Lu et al., 2022; Khan et al., 2024), interactions are organized into trials. Each trial comprises $N$ parallel environments, where agents engage in $E$ episodes, each comprising $T$ rounds of the specified matrix game (Figure 1 provides a schematic representation of a trial).

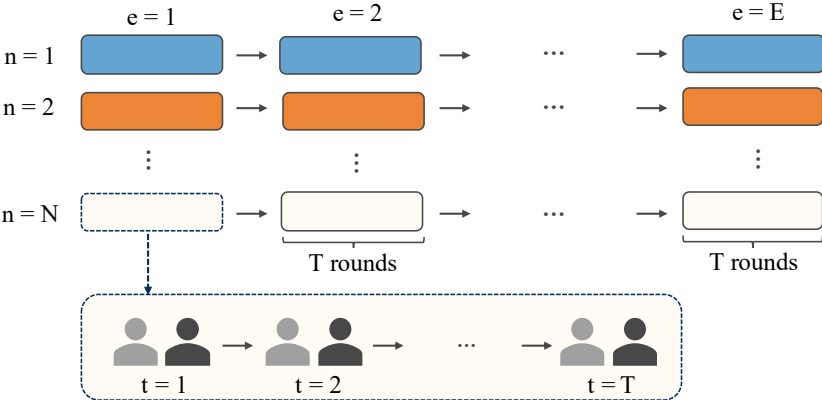

Figure 1: Schematic representation of a trial. Each box corresponds to an episode (a game played for $T$ rounds). Same-colored boxes represent episodes within the same parallel environment. Within each environment, episodes occur sequentially as indicated by the arrows. The shaper updates its parameters using the experience collected throughout the entire trial.

Let $t \in \{1, \ldots, T\}$ denote the round within an episode, and $e \in \{1, \ldots, E\}$ denote the episode within a trial. For notational simplicity, we collapse the pair $(e, t)$ into a single timescale index $\tau \in \{1, \ldots, E \times T\}$. We formalize the shaping task as a POMDP $(\bar{\mathcal{S}}, \bar{\mathcal{A}}, \bar{\mathcal{P}}, \bar{\mathcal{R}}, \bar{\Omega}, \bar{\mathcal{O}}, \bar{\gamma})$. The state $\bar{s}^\tau = \{\theta_i^{\tau-1}, c_i^{\tau-1}\}_{i \in I} \in \bar{\mathcal{S}}$ encodes the parameters and conditioning prompts of all LLM agents from the previous step. The action space $\bar{\mathcal{A}}$ and reward function $\bar{\mathcal{R}}$ are equivalent to those of the underlying repeated normal-form game. The observation $\bar{o}^\tau = f(\mathbf{a}^1, \mathbf{a}^2, \ldots, \mathbf{a}^{\tau-1}) \in \bar{\Omega}$ is a function of the joint actions across all past steps. Lastly, $\bar{\mathcal{P}}$ and $\bar{\mathcal{O}}$ denote the state transition and observation functions, respectively, and $\bar{\gamma}$ denotes the discount factor.

At the beginning of training, players are initialized with policies $\{\rho_{\theta_i}^0\}_{i \in I}$ and receive initial observations $\{c_i^0\}_{i \in I}$ specifying the game characteristics (number of players, action space, and reward matrix) and action labels. At the $\tau$-th round of a trial, the shaper receives an observation $c_j^\tau$ that concatenates two components: the most recent joint action $(\mathbf{a}^{\tau-1})$ and a compressed natural

language representation of all the previous joint actions in the trial ($f(\mathbf{a}^1, \ldots, \mathbf{a}^{\tau-2})$). This separation captures the distinction between history and context. The shaper then samples an action $a_j^\tau \sim \rho_{\theta_j}(w \mid c_j^\tau)$ and receives the corresponding reward $r_j^\tau$ and next observation $c_j^{\tau+1}$.

The opponent update their policy parameters at the end of each episode to maximize their return $J_i = \sum_{t=1}^{T} r_i^t$, where $r_i^t$ is the reward obtained in round $t$ by player $i$ from the game's payoff matrix (Appendix A.1). Consequently, within each trial, the shaper is exposed to $E$ opponent updates, though only indirectly through the evolving summaries of joint actions that persist across episodes. It is worth noting that opponent parameters are not reset between trials. By contrast, the shaper's own parameters $\theta_j$ are updated only at trial finalization to maximize the cumulative trial return $\bar{J}_j = \sum_{e=1}^{E} J_j^e = \sum_{e=1}^{E} \sum_{t=1}^{T} r_j^{(e,t)} = \sum_{\tau=1}^{E \times T} r_j^\tau$, where $r_j^{(e,t)}$ denotes the shaper's reward in round $t$ of episode $e$. Crucially, both agents receive rewards from the same payoff matrix. While we present this formulation in the context of repeated normal-form games, the ShapeLLM framework generalizes to any environment that can be formulated as a partially observable stochastic game.

## 4 EXPERIMENTAL SETTINGS

### 4.1 ENVIRONMENTS

We investigate opponent shaping on iterated versions of four canonical $2 \times 2$ games. These environments were selected to represent diverse incentive structures across strategic interactions.

**Iterated Prisoner's Dilemma (IPD)**. Players choose between cooperation (C) and defection (D). Mutual cooperation yields the highest collective payoff, but each player faces individual incentives to defect and exploit cooperative opponents (Rapoport, 1974; Axelrod & Hamilton, 1981).

**Iterated Matching Pennies (IMP)**. A zero-sum game where players choose between heads (H) and tails (T). One player receives a positive payoff when actions match, whereas the other is rewarded when they differ. This environment is purely adversarial.

**Iterated Chicken Game (ICG)**. Players can either Swerve (S) or Go straight (G). Going straight yields an advantage against a swerving opponent, but mutual aggression results in catastrophic outcomes for both, creating a coordination problem under risk (Rapoport & Chammah, 1966).

**Iterated Stag Hunt (ISH)**. Players choose between Stag (S) and Hare (H). Hunting stag yields the highest payoff but only if both players coordinate, while hunting hare offers a lower but guaranteed reward. This creates a coordination problem with multiple equilibria.

We assign a single token $w_{a_i}$ to each action $a_i \in \mathcal{A}$ (e.g. "C" for cooperate and "D" for defect in the IPD), and treat any other generation as an illegal action ($a_{\text{null}}$). Choosing $a_{\text{null}}$ incurs a penalty $r_{\text{null}}$, and the transition is excluded from both players' game histories (see Appendix A.2).

### 4.2 IMPLEMENTATION DETAILS

Our base model is *gemma-2-2b-it* (Gemma Team, 2024), a small, instruction-tuned, open-source language model. We focus on small models for computational efficiency and choose instruction-tuned variants as they are more goal-directed (Ouyang et al., 2022) and benefit from coding data exposure (Duan et al., 2024). To keep the agent architecture minimal, we restrict memory to the context window of the model and avoid additional reasoning scaffolds such as chain-of-thought (CoT) prompting (Wei et al., 2022b), which are less effective in small models (Wei et al., 2022a).

We train our agents using QLoRA (Dettmers et al., 2023), with the base model quantised to 4-bit precision via the `BitsAndBytes` package (Dettmers & von Koeller, 2022), and adapters of rank $r = 2$ implemented through the `PEFT` library (Mangrulkar et al., 2022). The learnable parameters comprise the LoRA adapters for the query/value projections and the value head parameters. All models are fine-tuned using a custom implementation of PPO that inherits from the `TRL` package[1] (von Werra et al., 2020)[2]. We run PPO training for 200-300 trials, with $N = 5$ parallel environments, $E = 5$ episodes, and $T = 20$ rounds per episode. For the shapers, we express context via cumulative

---

[1]The default implementation is only compatible with contextual bandits.
[2]Code available at `https://github.com/martaemili/shape-llm`

state visitation counts[3] (e.g., in the IPD: "CC: 1, CD: 1, DC: 2, DD: 3"). All training was done on a single A100 GPU with 40G of VRAM. The full specification of the hyperparameters, reward matrices, and training prompts used is provided in Appendices A.3, A.1, A.11, respectively.

## 5 SHAPING IN EXPLOITATIVE SETTINGS

We consider two core training configurations for our agents across the IPD, IMP and ICG.

**Baseline**. We establish the baseline performance using two LLM-based *naive learners* (NL) that treat their opponent as a stationary component of the environment. Each agent's conditioning prompt contains the game description and the most recent joint action. Once the episode is finished, both players simultaneously update their parameters via PPO to maximize episodic returns. This baseline establishes expected behavior when no opponent shaping occurs.

**Shaper vs. naive learner**. An LLM-based shaper interacts with an LLM naive learner (with the same configuration as in the baseline). As described in Section 3.2, the shaper updates its parameters only at trial completion, after having observed multiple opponent parameter updates.

Following training, we evaluate performance by having each trained pair of agents play 100 games with the same episode length used during training ($T = 20$).

### 5.1 SHAPING IN THE IPD, IMP AND ICG

We run experiments across 5 random seeds[4] using action labels $w_{a_1} = $ C, $w_{a_2} = $ D for the IPD, $w_{a_1} = $ H, $w_{a_2} = $ T for the IMP, and $w_{a_1} = $ S, $w_{a_2} = $ G for the ICG. Figure 2 illustrates the training dynamics for the shaper experiments across the three games (the corresponding figures for the baselines are shown in Appendix A.10).

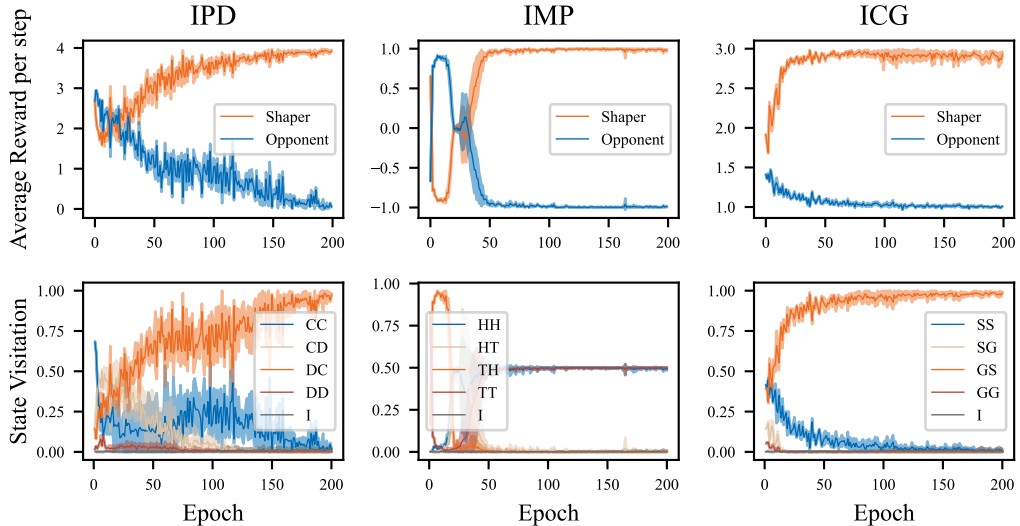

Figure 2: Average reward per step (top row) and state visitation (bottom row) during training for the shaping experiments across the IPD, IMP, and ICG. In the state visitation figures, the outcome "I" encompasses all transitions where either player chose $a_{\text{null}}$. Results are reported along with a 95% confidence interval over 5 random seeds.

---

[3]We represent the context via visitation counts instead of full trajectories to prevent the token length from growing linearly with the number of rounds.

[4]For the ICG baseline, we conducted experiments over 10 seeds. Each seed converges to one of the two possible Nash Equilibria, so a larger sample size is needed to accurately estimate the percentage of convergence to each equilibrium.

Table 1 presents the post-training evaluation results, where each jointly-trained pair played 100 games with episode length $T = 20$. The results demonstrate successful opponent shaping across all environments. For completeness, we evaluate performance across games of varying lengths (see Appendix A.7) and observe a similar performance.

Table 1: Post-training evaluation results for the IPD, IMP, and ICG comparing baseline (two naive learners) versus shaper-naive learner pairs. Average rewards per step are reported with 95% confidence intervals across 5 random seeds, except for the ICG baseline, where we use 10. Transitions involving $a_{\text{null}}$ are excluded (comprising 2% of actions in IPD, 0.1% in IMP, and 1% in ICG).

|     | **Baseline** | | **One Shaper** | |
| --- | --- | --- | --- | --- |
|     | Player 1 | Player 2 | Shaper | Opponent |
| IPD | $1.00 \pm 0.00$ | $1.00 \pm 0.00$ | $3.96 \pm 0.01$ | $0.10 \pm 0.04$ |
| IMP | $-0.03 \pm 0.09$ | $0.03 \pm 0.09$ | $0.99 \pm 0.01$ | $-0.99 \pm 0.01$ |
| ICG | $2.00 \pm 0.58$ | $2.00 \pm 0.58$ | $2.98 \pm 0.01$ | $1.01 \pm 0.01$ |

We begin by examining the performance in the IPD. In the baseline, both learners converge to mutual defection, which is the Nash Equilibrium, achieving an average payoff of 1. In contrast, the shaper achieves an average reward of 3.96, exceeding what any zero-determinant extortion (Press & Dyson, 2012) or tit-for-tat strategy could obtain. Meanwhile, the opponent achieves 0.1, which is lower than the mutual defection payoff. The training dynamics show a three-phase pattern: starting from high initial cooperation, the shaper first sharply reduces its cooperation rate, then plateaus at a stable level to maintain opponent cooperation, and finally slowly decreases cooperation to achieve near-maximal exploitation.

We observe similar performance patterns in the ICG. The baseline results show average rewards of 2 for both players. The standard deviations of 0.58 reflect the fact that each seed converges to one of the two pure Nash equilibria: either (Swerve, Go Straight) or (Go Straight, Swerve). In contrast, the shaper consistently achieves an average reward of 2.98 while limiting its opponent to 1.01. The training dynamics differ from the IPD: the shaper adopts an aggressive strategy by sharply reducing its swerving probability early in training, forcing convergence to its preferred equilibrium.

Finally, examining the IMP results, we observe that both agents in the baseline oscillate around the mixed Nash equilibrium with near-zero average payoffs (-0.03 and 0.03 respectively). In contrast, when one agent is a shaper, we observe clear exploitation, with the shaper obtaining a reward of 0.99 while the opponent obtains -0.99. The state visitation converges to equal frequency for the two states that favor the shaper: (H, H) and (T, T).

Collectively, these results demonstrate that the shaper consistently outperforms its opponent across all games, providing strong evidence of its ability to successfully influence opponents' learning dynamics in adversarial and mixed-motive scenarios. To establish the robustness of these results, we conduct several additional experiments. First, we conduct an ablation study to determine whether shaping effects stem solely from the shaper's enriched observation space (Appendix A.4). We adopt the same settings as in the baseline experiments, with one of the agents receiving a summary of all interactions in the current episode. Second, we examine sensitivity to prompt variations by running shaping experiments with actions presented in reversed order and an alternative prompt formulation (Appendix A.5). These experiments confirm that ShapeLLM achieves robust shaping across different configurations, and that enriched observations alone are insufficient for effective opponent shaping. Third, to gain better insights into the shaping mechanism, we provide an ablation study in Appendix A.8 in which we remove both the intra- and inter-episode history, and the inter-episode history only, from the shaper's observations in the IPD. These confirm that both intra- and inter-episode information are essential for effective shaping. Finally, to assess cross-model generalization, we explore the baseline and shaper experiments for the IPD using Llama-3.2-1B-Instruct Llama Team, AI@Meta (2024b) as a base model in Appendix A.9.

## 5.2 Robustness against Different Opponents

A robust shaping procedure should be capable of successfully influencing the learning dynamics of a diverse set of opponents. To test this capability, we explore shaping against opponents with distinct

initial policies. For each game, we systematically select three action label pairs yielding initial probabilities of playing action $a_1$[5] of 0.75, 0.5, and 0.25. We then conduct shaping experiments using these labels for the opponents. A detailed description of the selection method, action labels, and initial output probabilities for each opponent is provided in Appendix A.6. We train shapers against each selected opponent using the same procedure as in Section 5.1, with evaluation results shown in Table 2.

Table 2: Post-training evaluation results for the shaping experiments in the IPD, IMP and ICG across different opponent initializations. Each column represents a distinct opponent characterized by its approximate initial probability of playing action $a_1$. Average rewards per step are reported with 95% confidence intervals across 5 seeds. Transitions where $a_{\text{null}}$ is played by either player are excluded from the analysis (comprising 0-2% of all transitions).

| | $p_{\text{NL}}^0(a_1) \sim 0.75$ | | $p_{\text{NL}}^0(a_1) \sim 0.50$ | | $p_{\text{NL}}^0(a_1) \sim 0.25$ | |
| | Shaper | Opponent | Shaper | Opponent | Shaper | Opponent |
|---|---|---|---|---|---|---|
| IPD | $3.99 \pm 0.01$ | $0.01 \pm 0.02$ | $3.95 \pm 0.01$ | $0.04 \pm 0.03$ | $3.98 \pm 0.02$ | $0.07 \pm 0.07$ |
| IMP | $0.96 \pm 0.02$ | $-0.96 \pm 0.02$ | $0.99 \pm 0.01$ | $-0.99 \pm 0.01$ | $0.99 \pm 0.01$ | $-0.99 \pm 0.01$ |
| ICG | $3.00 \pm 0.00$ | $1.00 \pm 0.00$ | $2.99 \pm 0.01$ | $1.01 \pm 0.01$ | $2.95 \pm 0.01$ | $1.05 \pm 0.01$ |

Across all games and opponent types, shapers successfully exploit their co-players, achieving average per-step rewards of 3.97 in the IPD, 0.98 in the IMP, and 2.98 in the ICG. In the IPD and ICG, opponents converge to less favorable outcomes when initialized with more cooperative policies: their average rewards per step range from 0.01 to 0.07 in the IPD and from 1.00 to 1.05 in the ICG as initial policies become increasingly defective. The training dynamics (Appendix A.10) reveal that shapers respond strategically to opponent initialization. Against more cooperative opponents, shapers reduce their own cooperation more rapidly and reach lower final cooperation levels. This pattern indicates that initially defective agents require more prolonged cooperation incentives before they can be effectively exploited.

In contrast, the IMP shows no sensitivity to different opponent initializations. Intuitively, shaping should be more challenging against initial policies closer to the mixed Nash equilibrium. However, our results reflect no such effect. It is worth noting that the opponent with $p_{\text{NL}}^0 \sim 0.5$ is not initialized with a purely random policy, but with the closest approximation achievable through action label selection (see Appendix A.6 for the exact initial policy). Shapers converge to near-optimal outcomes, achieving rewards of 0.96, 0.99, and 0.99.

## 6 SHAPING IN COOPERATIVE SETTINGS

In the ISH baselines both agents achieve a mean reward of 1.30, where 90% of runs converge to the Pareto-inferior equilibrium (both hunt Hare)[7]. With a shaper present, runs consistently converge to the Pareto-optimal equilibrium (both hunt Stag), both achieving rewards of approximately 3.96. This shows that shaping can resolve coordination failures and guide systems toward mutually beneficial outcomes when cooperation is required. In the cooperative IPD variant, baseline runs converge to the Nash equilibrium (mutual defection) with rewards of 1 each. With a shaper, all runs achieve mutual cooperation, yielding rewards of 5.88 and 2.86 for the shaper and naive learner, respectively. This outcome demonstrates that shaping can achieve globally beneficial outcomes in environments where other players have mixed incentives.

---

[5]The action $a_1$ corresponds to cooperation in the IPD, playing heads in the IMP, and swerving in the ICG.

[6]We considered a variant where the shaper received the sum of both players' payoff as a reward. This configuration shifted the Nash equilibrium to an asymmetric outcome where the shaper cooperates while the opponent defects.

[7]We run the ISH and C-IPD baseline experiments across 10 seeds.

We investigate whether shaping can guide interactions toward mutually beneficial rather than purely exploitative outcomes. We explore this in two environments: a cooperative variation of the IPD (C-IPD) and the Iterated Stag Hunt (ISH). For the C-IPD, we provide both agents with the original payoff matrix, but modify the shaper's version so that its highest payoff is achieved through mutual cooperation, with all other payoffs unchanged[6]. This could be interpreted as the shaper receiving an intrinsic reward when the most globally beneficial outcome is achieved. The reward matrices used can be found in Appendix A.1. We use action labels $w_{a_1} = \text{C}, w_{a_2} = \text{D}$ for the C-IPD, and $w_{a_1} = \text{S}, w_{a_2} = \text{H}$ for the ISH. Figure 3 illustrates the training dynamics for the shaper experiments across the two environments (the corresponding figures for the baselines are shown in Appendix A.10). Table 3 presents the post-training evaluation results.

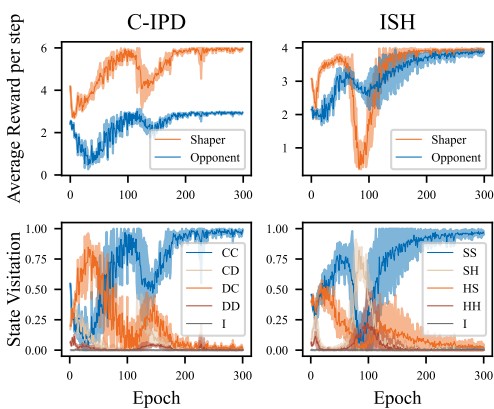

Figure 3: Average reward per step (top row) and state visitation (bottom row) during training for the shaping experiments across the C-IPD and ISH. In the state visitation figures, the outcome "I" encompasses all transitions where either player chose $a_{\text{null}}$. All results are reported along with a 95% confidence interval over 5 random seeds.

Table 3: Post-training evaluation results for the C-IPD and ISH comparing baseline versus shaper-naive learner pairs. Average rewards per step are reported with 95% confidence intervals across 5 and 10 random seeds for shaping and baseline experiments, respectively. Transitions with $a_{\text{null}}$ are excluded from the analysis ($\sim 2\%$ and $\sim 0.1\%$ of actions, respectively).

| | Baseline | | One Shaper | |
|---|---|---|---|---|
| | Player 1 | Player 2 | Shaper | Opponent |
| C-IPD | $1.00 \pm 0.00$ | $1.00 \pm 0.00$ | $5.88 \pm 0.03$ | $2.86 \pm 0.02$ |
| ISH | $1.30 \pm 0.52$ | $1.30 \pm 0.52$ | $3.96 \pm 0.02$ | $3.96 \pm 0.02$ |

## 7 DISCUSSION

**Implications.** In this work, we have demonstrated that LLM-based agents can be susceptible to opponent shaping in both exploitative and cooperative game-theoretic settings. As LLM agents become increasingly deployed in real-world applications, they will inevitably interact with other agents, and potentially train on data acquired through these interactions (for example, in the case of continually learning LLMs). In such settings, our findings suggest that agents could be vulnerable to strategic exploitation by opponents with no knowledge or control over them. Conversely, the same mechanisms could be leveraged beneficially, enabling agents to guide interactions toward mutually beneficial outcomes regardless of the goals of their opponents.

**Limitations.** Our work has several limitations that suggest promising directions for future research. First, due to computational constraints, we have only evaluated our approach using a single small model (*gemma-2-2b-it*). Future work could investigate whether shaping capabilities generalize to larger models and explore the relationship between model scale and shaping dynamics. For instance, whether smaller models are more vulnerable to these influences or whether larger models possess enhanced shaping capabilities. Second, our experiments focus solely on interactions among LLM agents. Future work could investigate cross-architecture shaping dynamics, examining whether LLM-based shapers can influence the learning dynamics of other types of agents or vice versa. Third, our experiments instructed agents to select from a fixed set of action tokens. While this restriction made evaluation tractable, it limits the ways in which LLMs can influence one another. In practice, LLM agents can communicate through natural language, and expanding interactions beyond fixed tokens may substantially alter shaping dynamics. Even within the same game-theoretic

settings, agents could employ language strategically, for example, by signaling intentions or negotiating before making a move. Future work could examine whether such natural language interaction strengthens, weakens, or qualitatively changes shaping outcomes. Finally, our study was restricted to $2 \times 2$ matrix games, where incentives are unambiguous and easily interpreted. Many real-world interactions, however, involve more nuanced or overlapping objectives, where cooperation and competition are not strictly binary. Exploring shaping in environments with richer payoff structures or multiple objectives would yield a deeper understanding of how these dynamics generalize to more realistic settings.

## 8 CONCLUSION

In this paper, we have investigated whether opponent shaping, a well-established technique in multi-agent reinforcement learning, extends to LLM-based agents. To the best of our knowledge, this is the first work to study opponent shaping with LLM agents. We proposed ShapeLLM, a model-free opponent shaping method for transformer-based agents, and demonstrated successful shaping in both exploitative and cooperative settings. In exploitative scenarios, LLM shapers influenced opponent learning in repeated games such as the IPD, IMP, and ICG, steering convergence toward outcomes that maximized their own payoff. In cooperative scenarios, shaping promoted coordination in settings like the ISH and a modified IPD, guiding agents toward mutually beneficial equilibria. By demonstrating that LLMs can both shape and be shaped through interaction alone, our findings highlights the importance of understanding multi-agent dynamics when deploying these systems in shared environments.

## ACKNOWLEDGMENTS

This work was supported by the UK Engineering and Physical Sciences Research Council (EPSRC) through the Centre for Doctoral Training Studentship in Cybersecurity - EP/S022503/1 (Marta Emili Garcia Segura) and grant EP/X028569/1.

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

## A APPENDIX

### A.1 PAYOFF MATRICES AND ILLEGAL ACTION PENALTIES USED DURING TRAINING

In Section 5, we consider three environments: Iterated Prisoner's Dilemma (IPD), Iterated Matching Pennies (IMP), and the Iterated Chicken Game (ICG). Table 4 shows their corresponding payoff matrices, where each cell contains a tuple representing the payoffs of the row (first entry) and column (second entry) players. Actions are represented with the following labels: "C" for cooperate and "D" for defect in IPD, "H" for heads and "T" for tails in IMP, and "S" for swerve and "G" for go straight in ICG.

Table 4: Payoff matrices for the three environments considered in Section 5 to explore shaping in exploitative settings.

| | (a) IPD | | | (b) IMP | | | (c) ICG | |
|---|---|---|---|---|---|---|---|---|
| | C | D | | H | T | | S | G |
| C | (3, 3) | (0, 4) | H | (1, -1) | (-1, 1) | S | (2, 2) | (1, 3) |
| D | (4, 0) | (1, 1) | T | (-1, 1) | (1, -1) | G | (3, 1) | (-5, -5) |

In Section 6, we use opponent shaping to promote globally beneficial outcomes in two games: Iterated Stag Hunt (ISH) and a cooperative IPD variant (C-IPD). In the latter, one player receives an enhanced reward for mutual cooperation while all other payoffs remain unchanged from the standard IPD. Table 5 shows the corresponding payoff matrices, where C-IPD uses the same action labels as IPD, and ISH uses "S" for stag and "H" for hare.

Table 5: Payoff matrices for two environments considered in Section 6 to investigate shaping in cooperative settings.

|  | **(a) C-IPD** | | | **(b) ISH** | |
|---|---|---|---|---|---|
|  | C | D |  | S | H |
| C | (6, 3) | (0, 4) | S | (4, 4) | (0, 3) |
| D | (4, 0) | (1, 1) | H | (3, 0) | (1, 1) |

The penalty for generating an illegal action, $r_{\text{null}}$, is always set to one unit below the lowest reward in each game's payoff matrix. Specifically: $r_{\text{null}}^{\text{IPD}} = r_{\text{null}}^{\text{C-IPD}} = -1$, $r_{\text{null}}^{\text{IMP}} = -2$, $r_{\text{null}}^{\text{ICG}} = -6$ and $r_{\text{null}}^{\text{ISH}} = -1$.

## A.2 LLM GAMEPLAY IN REPEATED NORMAL-FORM GAMES

All the environments considered are 2×2 repeated normal-form games. Since our agents are LLM-based, even after restricting the generation length to one token, their output space is much larger than the game's action space ($|\mathcal{V}| >> |\mathcal{A}| = 2$).

To tackle this space mismatch, one could try shrinking the model's output space via logit masking or rejection sampling. However, these interventions can alter the masked logits in unexpected ways or lead to increased computational time. Instead of actively ignoring or hiding parts of the output space, we define a mapping $\phi : \mathcal{V} \to \mathcal{A}$.

Directly defining such a mapping would require distributing the entire vocabulary across two actions, resulting in semantically unrelated tokens being mapped to the same action. This would impose an unnecessary learning objective whereby agents must learn arbitrary semantic equivalences that are orthogonal to the underlying strategic objective. To avoid this, we introduce a null action $a_{\text{null}}$, such that $\mathcal{A}_i' = \mathcal{A}_i \cup \{a_{\text{null}}\}$. This action is not meant to represent refusal to engage in the game, but rather failure to produce a reasonable answer.

We then define $\phi_i : \mathcal{V} \to \mathcal{A}_i'$. This formulation is general and can accommodate open-ended generation if the mapping itself is another language model. For simplicity, we choose:

$$\phi_i(w) = \begin{cases} a_1 & \text{if } w = w_{a_1}, \\ a_2 & \text{if } w = w_{a_2}, \\ a_{\text{null}} & \text{otherwise}, \end{cases} \quad (2)$$

such that each action can be played with one specific token, while any other token is considered *illegal*. The generation is steered towards this format via textual instructions (e.g., *Reply with "C" or "D"*).

Augmenting the action space requires extending the payoff matrix. Table-6 shows the augmented payoff matrix for a general 2×2 matrix game. The revised matrix is identical to the original one when both players play legal actions. If an agent plays $a_{\text{null}}$, it receives a penalty $r_{\text{null}}$, regardless of its opponent's move. If the agent plays a legal action but its opponent does not, the transition is discarded.

With the augmented action space and payoff structure defined, we can now describe how agents interact within this framework. In the $t$-th round, each agent receives a context $c_i^t$, which is a sequence of tokens consisting of the game description and a summary of the previous rounds of the game. A single token is then sampled from the output distribution of each agent ($w_i^t \sim \rho_{\theta_i}(w|c_i^t)$) and subsequently mapped to a game action, such that $\mathbf{a}^t = \{a_i^t = \phi_i(w_i^t)\}_{i \in I}$. The environment then returns the corresponding rewards ($\{r_i^t = R_i(\mathbf{a}^t)\}_{i \in I}$) and contexts for the next round ($\{c_i^{t+1} = f(c_i^t, \mathbf{a}^t)\}_{i \in I}$).

Table 6: Augmented payoff matrix for training LLM-agents in repeated normal-form games. When both agents play legal actions, payoffs match those of the underlying game. When an agent plays an illegal action, it receives a penalty $r_{\text{null}}$, regardless of the opponent's action. However, when an agent plays a legal action but the opponent plays an illegal action, the transition is discarded as it provides no meaningful learning signal (indicated by dashes in the matrix).

|  | $\mathbf{a_1}$ | $\mathbf{a_2}$ | $\mathbf{a_{null}}$ |
|---|---|---|---|
| $\mathbf{a_1}$ | $r(a_1, a_1)$ | $r(a_1, a_2)$ | – |
| $\mathbf{a_2}$ | $r(a_2, a_1)$ | $r(a_2, a_2)$ | – |
| $\mathbf{a_{null}}$ | $r_{\text{null}}$ | $r_{\text{null}}$ | $r_{\text{null}}$ |

## A.3 TRAINING IMPLEMENTATION DETAILS FOR REPRODUCIBILITY

This section provides detailed hyperparameter specifications that supplement the implementation details in Section 4.2.

**Generation Parameters**. We use the same generation parameters for all agents across the training and evaluation phases. The configuration is kept to the default values with three exceptions: we enable sampling (`do_sample=True`), disable top-$k$ generation (`top_k=0`), and restrict the generation length to one token (`max_new_tokens=1`). All other parameters (e.g., `temperature=1.0`, `top_p=1.0`) remain at default values.

**Adapter Configuration.** We employ the same adapter configuration for all agents. We only modify the rank parameter (`r=2`) to reduce compute. All other parameters (e.g., `lora_alpha=32`, `lora_dropout=0.05`, `target_modules = ["q_proj", "v_proj"]`) are kept at their default values.

### A.3.1 NAIVE LEARNER HYPERPARAMETERS

We aimed to maintain hyperparameters at their default values for naive learners to simulate realistic scenarios where opponents cannot be controlled. However, several adjustments were necessary due to memory constraints and training instability. To reduce the memory consumption and compute, we reduced the adapter rank, batch size, and mini-batch size. These parameters were kept identical for both shapers and naive learners to ensure a fair comparison. Additionally, we observed that under the default settings some agents learned to generate a substantial amount of illegal actions. To counter this instability, we reduced the learning rate, the number of optimization epochs per batch (PPO epochs), and incorporated reward scaling to maintain stable training dynamics across all agents.

Table 7 presents the hyperparameters used for training naive learners across all experiments[8]. Parameters marked with the *"(default)"* flag indicate values that remained unchanged from the `TRL` library's default configuration.

### A.3.2 SHAPER HYPERPARAMETERS

For shapers, the hyperparameters used varied across games and opponents. However, several core parameters were held constant across all experiments, which are presented in Table 8. Where parameters deviate from default values, modifications were made either for computational efficiency (rank, batch size, mini batch size) or improved training stability (score scaling, PPO epochs)

We varied three main hyperparameters across experiments: the learning rate (*lr*), the value function coefficient ($c_{\text{VF}}$) and the clipping range ($\epsilon_{\text{p}}$). The learning rate and clipping range were reduced mainly to increase stability during training. Under the default settings, we observed high variation in convergence outcomes across different random seeds. The value function coefficient $c_{\text{VF}}$ weights the value function term in the PPO loss. Shapers operate over much longer horizons than naive learners, requiring them to predict expected returns at the trial level rather than the episode level. This creates a challenging value prediction problem: the value function must estimate returns ranging from entire

---

[8]These parameters were used for all naive learners and games except IMP. Under these parameters, naive learners playing the IMP converged to deterministic strategies rather than the expected mixed Nash equilibrium. We therefore reduced the learning rate to $1.41 \times 10^{-7}$ and the value function coefficient to 0.05 for all IMP naive learners.

Table 7: Naive learner hyperparameters used in all experiments. The *"(default)"* flag indicates hyperparameters taking the default value in the `TRL` package.

| Parameter | Value |
|---|---|
| LoRA rank | 2 |
| LoRA target modules | ["q_proj", "v_proj"] (default) |
| Learning rate | $1.41 \times 10^{-6}$ |
| Use adaptive KL control | Yes (default) |
| Starting KL coefficient | 0.2 (default) |
| Target KL value | 6.0 (default) |
| Horizon for adaptive KL control | 10000 (default) |
| GAE $\gamma$ | 1.0 (default) |
| GAE $\lambda$ | 0.95 (default) |
| Clipping range | 0.2 (default) |
| Value Function clipping | 0.2 (default) |
| Value Function Loss coefficient | 0.2 (default) |
| Batch Size | 100 |
| Mini Batch Size | 10 |
| Gradient Accumulation Steps | 1 (default) |
| PPO epochs | 1 |
| Score normalization | No (default) |
| Score scaling | Yes |

Table 8: Shaper hyperparameters fixed across all experiments. The *"(default)"* flag indicates hyperparameters taking the default value in the `TRL` package.

| Parameter | Value |
|---|---|
| LoRA rank | 2 |
| LoRA target modules | ["q_proj", "v_proj"] (default) |
| Use adaptive KL control | Yes (default) |
| Starting KL coefficient | 0.2 (default) |
| Target KL value | 6.0 (default) |
| Horizon for adaptive KL control | 10000 (default) |
| GAE $\gamma$ | 1.0 (default) |
| GAE $\lambda$ | 0.95 (default) |
| Value Function clipping | 0.2 (default) |
| Batch Size | 100 |
| Mini Batch Size | 10 |
| Gradient Accumulation Steps | 1 (default) |
| PPO epochs | 1 |
| Score normalization | No (default) |
| Score scaling | Yes |

trials (for initial states) to single immediate rewards (for final states). Since the shaper's value head is randomly initialized, under the default $c_{\text{VF}}$ value, the initial value loss dominates the policy loss by orders of magnitude, leading to large gradients and causing training instability. A particularly problematic consequence is that agents sometimes learn to generate illegal tokens simply to make the value prediction task easier. We address this issue by reducing the value function coefficient to better balance the relative contributions of the value and policy losses.

Table 9 shows the learning rate, value function coefficient, and clipping range used to train the shapers across all experiments in the main text. The remaining hyperparameters used during training for all experiments are shown in Table 8.

Table 9: Shaper's learning rate ($lr$), value function coefficient ($c_{VF}$), and clipping range ($\epsilon_p$) for the experiments in Sections 5.1, 5.2, 6.

|  | Experiment | $lr$ | $c_{VF}$ | $\epsilon_p$ |
|---|---|---|---|---|
| Section 5.1 | IPD | $1.41 \times 10^{-7}$ | $10^{-3}$ | $10^{-4}$ |
|  | IMP | $3.41 \times 10^{-7}$ | $10^{-3}$ | $2 \times 10^{-1}$ |
|  | ICG | $1.41 \times 10^{-7}$ | $10^{-3}$ | $2 \times 10^{-1}$ |
| Section 5.2 | IPD with $p^0_{NL}(a_1) \sim 0.75$ | $1.41 \times 10^{-7}$ | $10^{-3}$ | $2 \times 10^{-1}$ |
|  | IPD with $p^0_{NL}(a_1) \sim 0.5$ | $1.41 \times 10^{-7}$ | $5 \times 10^{-4}$ | $5 \times 10^{-3}$ |
|  | IPD with $p^0_{NL}(a_1) \sim 0.25$ | $1.41 \times 10^{-7}$ | $3 \times 10^{-3}$ | $10^{-4}$ |
|  | IMP with $p^0_{NL}(a_1) \sim 0.75$ | $4.41 \times 10^{-7}$ | $10^{-3}$ | $2 \times 10^{-1}$ |
|  | IMP with $p^0_{NL}(a_1) \sim 0.5$ | $4.41 \times 10^{-7}$ | $10^{-3}$ | $2 \times 10^{-1}$ |
|  | IMP with $p^0_{NL}(a_1) \sim 0.25$ | $6.41 \times 10^{-7}$ | $10^{-3}$ | $2 \times 10^{-1}$ |
|  | ICG with $p^0_{NL}(a_1) \sim 0.75$ | $1.41 \times 10^{-7}$ | $10^{-3}$ | $2 \times 10^{-1}$ |
|  | ICG with $p^0_{NL}(a_1) \sim 0.5$ | $1.41 \times 10^{-7}$ | $10^{-3}$ | $2 \times 10^{-1}$ |
|  | ICG with $p^0_{NL}(a_1) \sim 0.25$ | $6.41 \times 10^{-8}$ | $10^{-3}$ | $2 \times 10^{-1}$ |
| Section 6 | C-IPD | $8.41 \times 10^{-8}$ | $5 \times 10^{-5}$ | $2 \times 10^{-1}$ |
|  | ISH | $8.41 \times 10^{-8}$ | $10^{-3}$ | $2 \times 10^{-1}$ |

### A.3.3 PACKAGE VERSIONS

We used the following versions for the main Python packages:

- `BitsAndBytes`: v0.45.0
- `PEFT`: v0.14.0
- `torch`: v2.5.1
- `transformers`: v4.47.0
- `TRL`: v0.11.4

## A.4 ABLATION STUDY: NAIVE LEARNERS WITH ENRICHED OBSERVATIONS

We conduct a variation of the baseline experiments to test whether the shaping effects we observe are solely a product of the shaper's enriched observation space. The experimental setup involves two naive learners, with one player's observation space augmented to include the state counts from all previous interactions in the current episode. Crucially, these observations are reset at episode boundaries (unlike the shaper, whose observations persist across episodes within a trial). To ensure comparability, we use the same action labels as in Section 5.1: $w_{a_1} = C, w_{a_2} = D$ for the IPD, $w_{a_1} = H, w_{a_2} = T$ for the IMP, and $w_{a_1} = S, w_{a_2} = G$ for the ICG. Both learners use the hyperparameters presented in Table 7.

Table 10: Post-training evaluation results for baselines where Player 1 receives enriched observations. Columns *ICG* and *ICG (alt. opp.)* correspond to experiments with varying action labels for Player 2 (*ICG* uses $w_{a_1} = S, w_{a_2} = G$, and *ICG (alt. opp.)* employs $w_{a_1} = N, w_{a_2} = M$). Average rewards per step are reported with 95% confidence intervals across 5 random seeds.

|  | IPD | IMP | ICG | ICG (alt. opp.) |
|---|---|---|---|---|
| Player 1 (enriched) | $1.00 \pm 0.00$ | $-0.05 \pm 0.09$ | $2.60 \pm 0.76$ | $1.00 \pm 0.00$ |
| Player 2 | $1.00 \pm 0.00$ | $0.05 \pm 0.09$ | $1.38 \pm 0.74$ | $3.00 \pm 0.00$ |

Table 10 reports the post-training average reward per step for the IPD, IMP, and ICG. In the IPD and IMP, despite the asymmetry in observations, outcomes mirror those in the standard baseline:

mutual defection (1 each) for the IPD, and oscillation around the Nash Equilibrium (-0.05 and 0.05) for the IMP. This equivalence to the baseline is not the same for the ICG. When both players use action labels $w_{a_1} = $ S, $w_{a_2} = $ G (row *ICG*), the player with enriched observations achieves 2.60, while its opponent receives only 1.38. As in the baseline, each run converges to one of the two pure Nash equilibria, but 80% of runs favor the equilibrium most beneficial to the player with enriched observations. At first glance, this suggests that the additional information, rather than training with ShapeLLM, could drive the shaping behavior observed in the ICG.

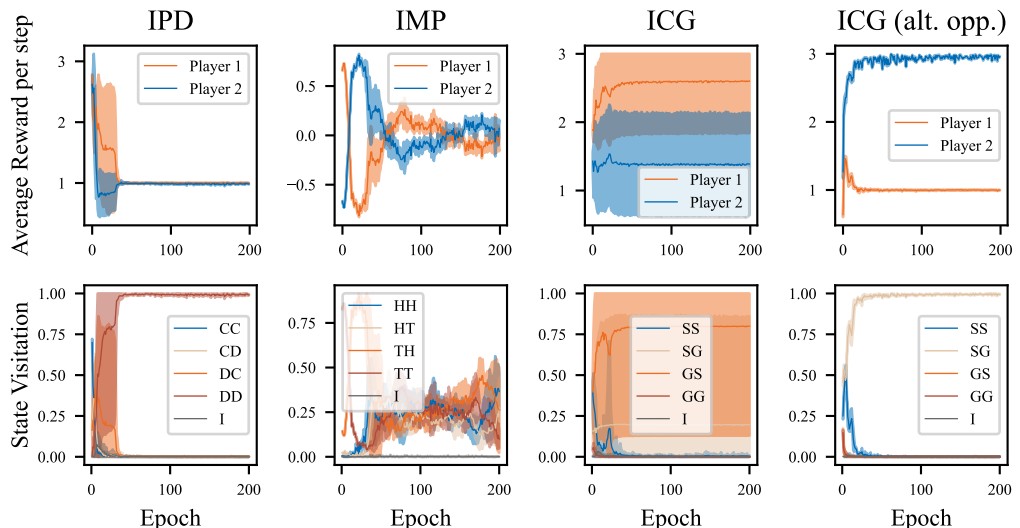

Figure 4: Average reward per step (top row) and state visitation (bottom row) during training for the enriched observation baseline experiments across the IPD, IMP, and ICG. For the latter, two opponent configurations are presented: *ICG* and *ICG (alt. opp.)*. They use $w_{a_1} = $ S, $w_{a_2} = $ G and $w_{a_1} = $ N, $w_{a_2} = $ M as the opponent's action labels respectively, and $w_{a_1} = $ S, $w_{a_2} = $ G for the player with enriched observations. In the state visitation figures, the outcome "I" encompasses all transitions where either player chose $a_{\text{null}}$. The results are reported along with a 95% confidence interval over 5 random seeds.

Inspecting the training dynamics (Figure 4), we observe a skewed initialization in the ICG, with 40% of transitions being (Go Straight, Swerve) at training initiation. To test whether the obtained results were a consequence of this initialization, we conduct a control experiment where Player 2 selects actions with labels $w_{a_1} = $ N, $w_{a_2} = $ M[9]. For this setup, the initial state visitation is starkly different (see Figure 4), with 46% of transitions being (Swerve, Go Straight). In this case, the enriched baseline for the ICG consistently converges to the least favorable equilibrium for Player 1 (with rewards of 1.00, 3.00 for Players 1 and 2, respectively).

This control shows that the apparent shaping advantage in the ICG enriched baseline was driven by favorable initialization, not by the enriched observation space itself. When the initialization is modified, the enriched learner systematically converges to the least favorable outcome. In contrast, when ShapeLLM is applied to the same opponent configuration, the shaper consistently achieves maximum rewards (see Section 5.2). Taken together, these results demonstrate that enriched observations alone are insufficient to produce shaping. Shaping requires the ability to indirectly observe and respond to the opponent's learning dynamics across episodes.

## A.5 ROBUSTNESS TO PROMPT VARIATIONS IN EXPLOITATIVE SHAPING EXPERIMENTS

We conduct two experiments for robustness to prompt variations. The first variation uses a prompt where the payoff matrix is presented in table form.

---

[9]These labels are used in Section 5.2 to show the shaping robustness under different opponent initializations.

---

**_Table-format base prompt_ for the IPD with $w_{a_1} = \mathbf{C}, w_{a_2} = \mathbf{D}$**

```
<bos><start_of_turn>user
You are playing a 2-player game with actions:  C, D. Points are assigned as
follows:

|       |  **C**  |  **D**  |
|-------|---------|---------|
| **C** |  (3, 3) |  (0, 4) |
| **D** |  (4, 0) |  (1, 1) |

Choose an action for the current round.  Reply only with C or D.
<end_of_turn>
<start_of_turn>model
```

Figure 5: Table-format prompt variation for the IPD. Instead of a textual description, the payoff matrix is presented in markdown table form, replicating the base model's formatting style.

h

Table 11: Post-training evaluation results for the IPD, IMP, and ICG with the table-format and switched-order prompts. Average rewards per step are reported with 95% confidence intervals across 5 random seeds. Transitions where $a_{\text{null}}$ is played by either player are excluded from the analysis (comprising 0-4% of all transitions).

| | Table-format | | Switched-order | |
|---|---|---|---|---|
| | Shaper | Opponent | Shaper | Opponent |
| IPD | $3.50 \pm 0.16$ | $0.53 \pm 0.09$ | $3.99 \pm 0.01$ | $0.02 \pm 0.01$ |
| IMP | $0.94 \pm 0.06$ | $-0.94 \pm 0.06$ | $0.98 \pm 0.04$ | $-0.98 \pm 0.04$ |
| ICG | $2.99 \pm 0.01$ | $1.01 \pm 0.01$ | $2.77 \pm 0.37$ | $1.23 \pm 0.37$ |

To determine the specific table formatting, we ask the base model to generate a general payoff matrix and replicate its output format (including spacing and header formatting). Figure 5 shows the _table-format base prompt_ used for the IPD.

For the IMP and ICG, the prompts are identical except for the modified action labels and payoff matrices. The example in Figure 5 shows the base prompt that both players receive at training initiation. The dynamic updates of this prompt throughout training (i.e., how history and context are updated) remain unchanged from those used in the main experiments (see Appendix A.11). We use the same action labels as in Section 5.1: $w_{a_1} = \text{C}, w_{a_2} = \text{D}$ for the IPD, $w_{a_1} = \text{H}, w_{a_2} = \text{T}$ for the IMP, and $w_{a_1} = \text{S}, w_{a_2} = \text{G}$ for the ICG. Figure 6, shows the training dynamics for the three games under the tabular prompt variation. It is worth noting that with this new prompt, both the initialization of the shaper and its opponent change significantly, especially for the IPD. Table 11 shows the evaluation results obtained.

The results demonstrate successful shaping across all three games. In the IPD, the shaper achieves 3.5 while its opponent receives 0.53, exceeding mutual cooperation payoffs but underperforming compared to the main text results. Since Section 5.2 shows consistent outcomes across different opponent initializations, this gap likely stems from the shaper's own initialization rather than opponent effects. The IMP shows a similar pattern with substantial initialization differences, with the shaper achieving 0.94 versus the opponent's -0.94. For the ICG, where initialization more closely matches the main setup, results are 2.99 for the shaper and 1.01 for the opponent.

For the second prompt variation, we reverse the order in which actions are presented (see Figure 7). As with the table-format experiments, the base prompts for IMP and ICG are identical except for their respective action labels and payoff matrices. We use the same action labels as in the previous variation.

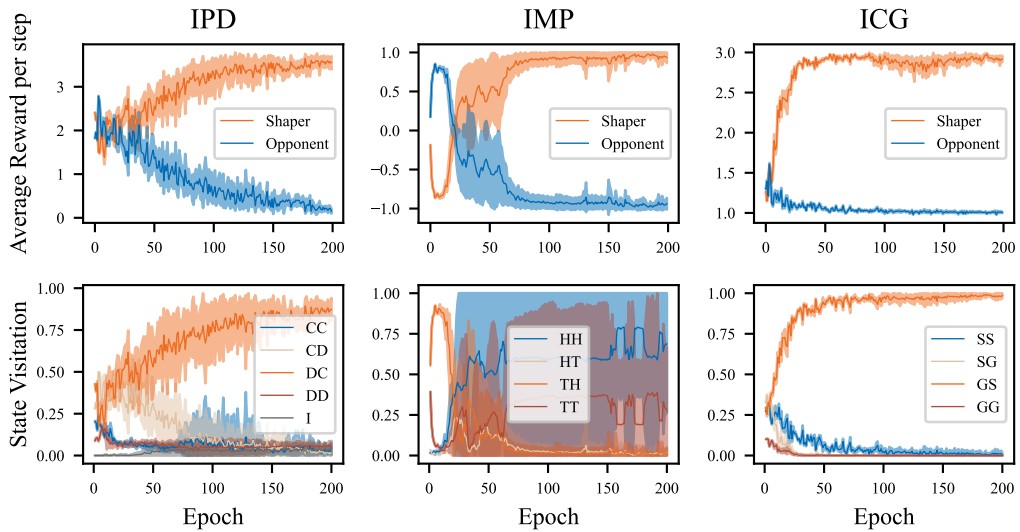

Figure 6: Average reward per step (top row) and state visitation (bottom row) during training for the shaping experiments with the table-format prompt across the IPD, IMP, and ICG. In the state visitation figures, the outcome "I" encompasses all transitions where either player chose $a_{\text{null}}$. Results are reported along with a 95% confidence interval over 5 random seeds.

---

**Switched-order base prompt** for the IPD with $w_{a_1} = \mathbf{C}, w_{a_2} = \mathbf{D}$

```
<bos><start_of_turn>user
You are playing a 2-player game with actions:  D, C. Points are assigned as
follows:  D/D: 1/1, D/C: 4/0, C/D: 0/4, C/C: 3/3.
Choose an action for the current round.  Reply only with C or D.
<end_of_turn>
<start_of_turn>model
```

Figure 7: Switched-order prompt variation for the IPD. The actions and payoffs are presented in reversed order compared to the main text prompt.

---

Figure 8 shows the training dynamics for the switched-order prompt experiments. As with the table-format variation, both agents exhibit substantially different initializations compared to the main experiments. Here, both players are heavily biased toward playing action $a_1$ (cooperate in the IPD, heads in the IMP, and swerving in the ICG), with their policies initially being almost deterministic (average of 99% of $(a_1, a_1)$ state at trial initiation). This creates challenges for the IPD and ICG, where this joint outcome yields acceptable rewards for both players. Consequently, without sufficient exploration incentives, the agents' policies remain unchanged for some of the seeds, focusing exclusively on the value prediction problem.

To address this issue, we introduce an entropy regularization term to the PPO loss function. While this term was included in the original PPO formulation (Schulman et al., 2017), it is not implemented in the `TRL` package. Since our agents' action space encompasses the entire vocabulary, maximizing their output distribution entropy would create conflicting incentives: illegal actions would incur penalties while simultaneously reducing the loss through increased entropy. Instead, we extract logits only for the two allowed action tokens and compute the entropy of the resulting normalized distribution. To prevent convergence to suboptimal policies, we employ a decaying entropy coefficient that gradually reduces the exploration incentive during training.

The switched-order prompt results are presented in Table 11. With entropy regularization applied to both the IPD and ICG, we again observe successful shaping with results closely matching those

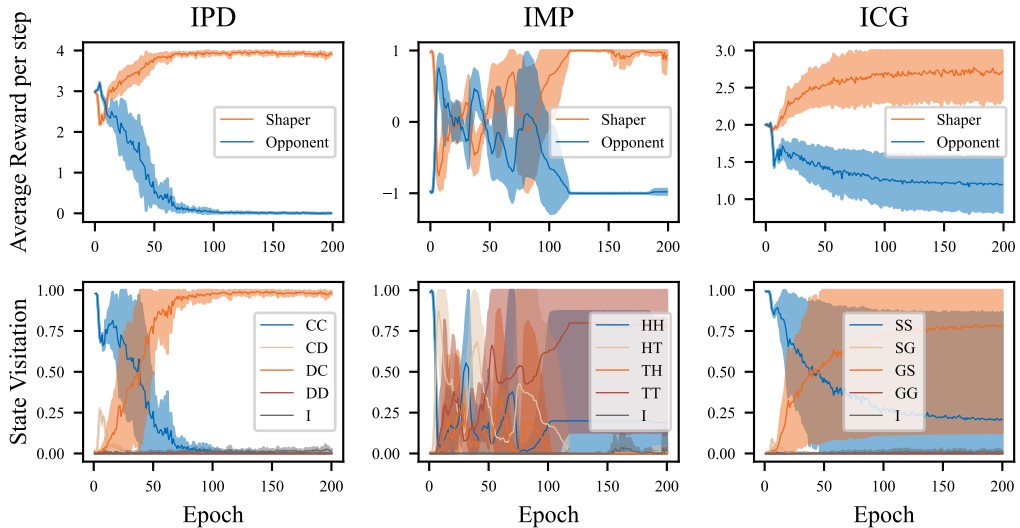

Figure 8: Average reward per step (top row) and state visitation (bottom row) during training for the shaping experiments with the switched-order prompt across the IPD, IMP, and ICG. In the state visitation figures, the outcome "I" encompasses all transitions where either player chose $a_{\text{null}}$. Results are reported along with a 95% confidence interval over 5 random seeds.

in the main text: 3.99 for IPD and 2.77 for ICG. In the ICG, while the shaper achieves an average reward of 3 across most runs, for one of the seeds it consistently fails to explore alternative actions, converging to the outcome where both players swerve. Lastly, the IMP performs similarly to other prompt formulations (0.94 for the shaper vs -0.94 for its opponent) without requiring an entropy regularization term.

These experiments demonstrate that ShapeLLM maintains robust shaping capabilities across different prompt formulations. Even when initial policies are nearly deterministic (as in the switched-order variation), introducing entropy regularization to encourage early exploration enables effective opponent shaping. For reproducibility, we list the hyperparameters modified in this appendix in Table 12. The rest of the hyperparameters used are specified in Table 9.

Table 12: Shaper's learning rate ($lr$), value function coefficient ($c_{\text{VF}}$), clipping range ($\epsilon_{\text{p}}$), and entropy regularization parameters ($c_{\text{S}}^{\text{init}}$, $c_{\text{S}}^{\text{end}}$, $T_{\text{S}}$) for the experiments in Appendix A.5.

| | Experiment | $lr$ | $c_{\text{VF}}$ | $\epsilon_{\text{p}}$ | $c_{\text{S}}^{\text{init}}$ | $c_{\text{S}}^{\text{end}}$ | $T_{\text{S}}$ |
|---|---|---|---|---|---|---|---|
| | IPD | $1.41 \times 10^{-7}$ | $10^{-3}$ | $10^{-5}$ | – | – | – |
| Table-format prompt | IMP | $2.41 \times 10^{-7}$ | $10^{-3}$ | $2 \times 10^{-1}$ | – | – | – |
| | ICG | $1.41 \times 10^{-7}$ | $10^{-3}$ | $2 \times 10^{-1}$ | – | – | – |
| | IPD | $6.41 \times 10^{-7}$ | $10^{-3}$ | $2 \times 10^{-1}$ | 0.1 | 0.0 | 25 |
| Switched-order prompt | IMP | $6.41 \times 10^{-6}$ | $10^{-3}$ | $2 \times 10^{-1}$ | – | – | – |
| | ICG | $1.41 \times 10^{-7}$ | $10^{-3}$ | $2 \times 10^{-1}$ | 0.7 | 0.0 | 25 |

## A.6    ACTION LABEL SELECTION FOR ROBUSTNESS EXPERIMENTS

LLM agent output distributions vary significantly with the choice of action labels ($w_{a_1}, w_{a_2}$). For the robustness experiments, we systematically selected action labels to achieve target initial output distributions for the opponents.

Table 13: Action labels that produce the closest output distributions to the target initial distributions for the IPD, IMP and ICG.

|  | $p^0_{NL}(a_1) \sim 0.75$ | | $p^0_{NL}(a_1) \sim 0.50$ | | $p^0_{NL}(a_1) \sim 0.25$ | |
|---|---|---|---|---|---|---|
|  | $w_{a_1}$ | $w_{a_2}$ | $w_{a_1}$ | $w_{a_2}$ | $w_{a_1}$ | $w_{a_2}$ |
| IPD | H | K | N | Y | I | X |
| IMP | S | Y | N | Y | N | M |
| ICG | T | K | N | M | T | F |

**Target Distributions**. For each game, we want to obtain opponents with initial probabilities of playing action $a_1$[10] of approximately 0.75, 0.5, and 0.25.

**Selection Procedure**. Opponents encounter 5 different prompts during training[11]. We extract the LLM's output probabilities for these 5 prompts across all possible combinations of capital letters as action labels (325 total combinations). For each combination, we calculate the KL divergence between the target and extracted output distributions, averaged across the 5 prompts, and select the combination with the lowest divergence. The resulting action labels are shown in Table 13.

The extracted initial probabilities for each pair of action labels in Table 13 are shown in Table 14, with separate results for the IPD (a), IMP (b), and ICG (c).

Table 14: Naive learner's initial action probabilities across three games (IPD, IMP, ICG) under varying action labels for the 5 distinct prompts encountered during training.

(a) Initial cooperation probability in the IPD.

| Action Labels | $p^0_{NL}(C)$ | $p^0_{NL}(C \mid CC)$ | $p^0_{NL}(C \mid CD)$ | $p^0_{NL}(C \mid DC)$ | $p^0_{NL}(C \mid DD)$ |
|---|---|---|---|---|---|
| $w_{a1} = H, w_{a2} = K$ | 0.60 | 0.89 | 0.89 | 0.70 | 0.68 |
| $w_{a1} = N, w_{a2} = Y$ | 0.68 | 0.38 | 0.87 | 0.58 | 0.77 |
| $w_{a1} = I, w_{a2} = X$ | 0.12 | 0.21 | 0.75 | 0.34 | 0.24 |

(b) Initial probability of playing heads in the IMP.

| Action Labels | $p^0_{NL}(H)$ | $p^0_{NL}(H \mid HH)$ | $p^0_{NL}(H \mid HT)$ | $p^0_{NL}(H \mid TH)$ | $p^0_{NL}(H \mid TT)$ |
|---|---|---|---|---|---|
| $w_{a1} = S, w_{a2} = Y$ | 0.71 | 0.78 | 0.87 | 0.57 | 0.70 |
| $w_{a1} = N, w_{a2} = Y$ | 0.48 | 0.32 | 0.60 | 0.56 | 0.66 |
| $w_{a1} = N, w_{a2} = M$ | 0.30 | 0.20 | 0.36 | 0.13 | 0.18 |

(c) Initial swerving probability in the ICG.

| Action Labels | $p^0_{NL}(S)$ | $p^0_{NL}(S \mid SS)$ | $p^0_{NL}(S \mid SG)$ | $p^0_{NL}(S \mid GS)$ | $p^0_{NL}(S \mid GG)$ |
|---|---|---|---|---|---|
| $w_{a1} = T, w_{a2} = K$ | 0.61 | 0.90 | 0.87 | 0.82 | 0.82 |
| $w_{a1} = N, w_{a2} = M$ | 0.41 | 0.40 | 0.71 | 0.37 | 0.61 |
| $w_{a1} = T, w_{a2} = F$ | 0.24 | 0.47 | 0.47 | 0.02 | 0.11 |

## A.7 Shaper evaluation for varying game lengths for the IPD, IMP, and ICG

Table 15 presents the evaluation results for the shapers trained in Section 5.1 for varying game lengths ($T = 50$, $T = 100$). The results demonstrate that shaper performance remains consistent

---

[10]The action $a_1$ corresponds to cooperation in the IPD, playing heads in the IMP, and swerving in the chicken game.

[11]One stateless prompt at training initiation, and 4 prompts corresponding to the 4 possible joint actions from the previous round (e.g., in the IPD: CC, CD, DC, DD).

across different game lengths, with no significant degradation in exploitation capability as episodes become longer.

Table 15: Post-training evaluation results for the IPD, IMP, and ICG comparing baseline (two naive learners) versus shaper-naive learner pairs for varying game lengths ($T = 20, 50, 100$). Average rewards per step are reported with 95% confidence intervals across 5 random seeds. Illegal actions are excluded from the analysis (comprising 2% of actions in IPD, 0.1% in IMP, and 1% in ICG).

| | **IPD** | | **IMP** | | **ICG** | |
|---|---|---|---|---|---|---|
| | Shaper | Opponent | Shaper | Opponent | Shaper | Opponent |
| $T = 20$ | $3.96 \pm 0.01$ | $0.10 \pm 0.04$ | $0.99 \pm 0.01$ | $-0.99 \pm 0.01$ | $2.98 \pm 0.01$ | $1.01 \pm 0.01$ |
| $T = 50$ | $3.97 \pm 0.01$ | $0.09 \pm 0.04$ | $0.99 \pm 0.01$ | $-0.99 \pm 0.01$ | $2.99 \pm 0.01$ | $1.01 \pm 0.01$ |
| $T = 100$ | $3.97 \pm 0.01$ | $0.09 \pm 0.04$ | $0.99 \pm 0.01$ | $-0.99 \pm 0.01$ | $2.99 \pm 0.00$ | $1.01 \pm 0.01$ |

### A.8 Ablation study: Varying intra- and inter-episode history in the IPD

We conduct two variations of our shaping experiments to test whether both intra- and inter-episode history are necessary for successful opponent shaping. Both experimental setups involve one shaper and one naive learner. In the first variation, the shaper receives only the current round's state, with no access to prior rounds or episodes. The shaper's parameters are still updated at the end of each trial, maintaining the asymmetric parameter update characteristic of ShapeLLM. In the second variation, we relax this constraint, with the shaper now receiving the full intra-episode history but no inter-episode information. In practice, this corresponds to the shaper's observations being reset at the end of each episode. Consequently, the shaper does not indirectly observe its opponent's parameter updates. The shaper's parameters are still updated only at the end of the trial. We test these variations in the IPD, with the hyperparameters used for the opponent and the shaper listed in Tables 7 and 9, respectively.

Table 16: Post-training evaluation results for the IPD with one shaper against a naive learner, where the shaper receives varying levels of intra- and inter-episode history. Average rewards per step are reported with 95% confidence intervals across 5 random seeds. Transitions with $a_{\text{null}}$ are excluded from the analysis ($\sim 1\%$ of actions).

| | **Current state only** | **Intra-episode history only** |
|---|---|---|
| Shaper | $0.99 \pm 0.00$ | $0.99 \pm 0.01$ |
| Opponent | $1.02 \pm 0.01$ | $1.02 \pm 0.02$ |

Table 16 reports the post-training average reward per step for the IPD when the shaper only receives the current state in its context, and when it receives intra-episode information only. In the first variation, we remove all intra- and inter-episode history from the shaper's context to test whether asymmetric parameter updates alone are sufficient for shaping to occur. In this setting, we obtain convergence to mutual defection, similarly to the baseline behavior, indicating that without additional history information shaping cannot occur despite asymmetric updates. For the second variation, in which intra-episode context is restored, we observe a similar pattern, with the configuration converging to mutual defection. This demonstrates that inter-episode information is essential for shaping, such that the shaper is able to indirectly observe its opponent's learning dynamics through context that persists across episodes.

Figure 9 shows the training dynamics for both ablation studies in the IPD. In both cases, the asymmetric parameter updates influence the learning dynamics. Unlike the baseline, where both agents quickly converge to mutual defection, here the shaper maintains a transient advantage over its opponent during early training. This advantage is more prolonged when intra-episode information is included, with agents maintaining higher cooperation rates, likely due to the in-context adaptation that additional history enables. However, the shaper ultimately fails to sustain opponent cooperation in both configurations, demonstrating that asymmetric parameter updates alone are insufficient to shape opponent learning dynamics.

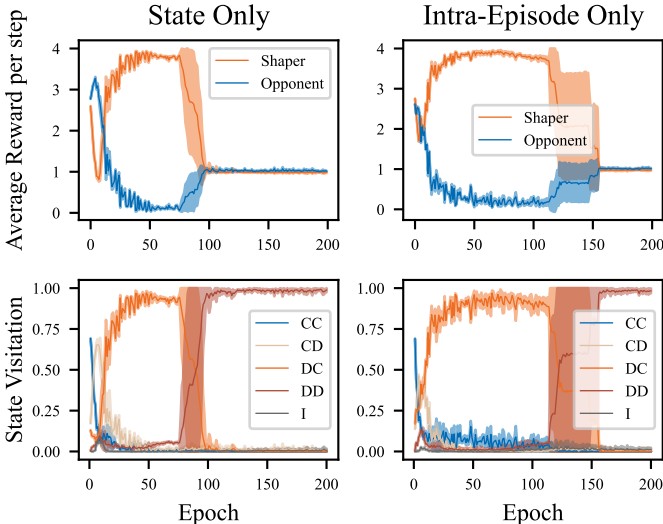

Figure 9: Average reward per step (top row) and state visitation (bottom row) during training for the IPD ablation experiments. The shaper receives either only the current state (left) or full intra-episode history (right), with no inter-episode information in either case. In the state visitation figures, the outcome "I" encompasses all transitions where either player chose $a_{\text{null}}$. The results are reported along with a 95% confidence interval over 5 random seeds.

## A.9   PRELIMINARY CROSS-MODEL VALIDATION OF SHAPELLM

ShapeLLM is designed to generalize across different LLM architectures, relying only on natural language context without any architecture-specific components. To provide preliminary evidence of cross-model generalization, we report baseline and shaper experiments for the IPD using *Llama-3.2-1B-Instruct* (Llama Team, AI@Meta, 2024b) as the base model.

The possible initial policies of *Llama-3.2-1B-Instruct* differ substantially from those of *gemma-2-2b-it*. As discussed in Section 5.2, *gemma-2-2b-it*'s initial policy for a given prompt depends on the choice of action labels. Figure 10 (a) shows heatmaps of *gemma-2-2b-it*'s initial output probabilities for tokens $w_{a_1}$ and $w_{a_2}$ across all 650 possible capital-letter action label pairs. Each heatmap corresponds to a different conditioning prompt (stateless or one of the four possible joint action states). Most policies concentrate in the diagonal (where $p(w_{a_1}|c)+p(w_{a_2}|c) \approx 1$), indicating a low probability of generating illegal actions. Additionally, despite the variation across action labels, policies for each conditioning prompt cluster in distinctive regions, indicating that the base model meaningfully differentiates between states.

In contrast, Figure 10 (b) shows that *Llama-3.2-1B-Instruct* produces substantially different behaviors. Nearly all policies lie off-diagonal, indicating 10-20% illegal action rates for most action label pairs. Moreover, unlike *gemma-2-2b-it*, policies do not cluster distinctly by conditioning prompt. The heatmaps for different states exhibit similar distributional patterns, suggesting the model does not meaningfully differentiate between game states based on previous joint actions.

To address these challenges and isolate architectural effects from differences in initialization, we perform supervised fine-tuning (SFT) to align *Llama-3.2-1B-Instruct*'s initial policy with that of *gemma-2-2b-it*. For the SFT procedure, we create a dataset of 1000 prompts that agents might encounter during IPD gameplay[12]. Each prompt has an associated target distribution, which cor-

---

[12]For each prompt, we uniformly sample an episode number, round number, and previous joint action. For state-occurrence prompts, we additionally sample state counts from a uniform multinomial distribution.

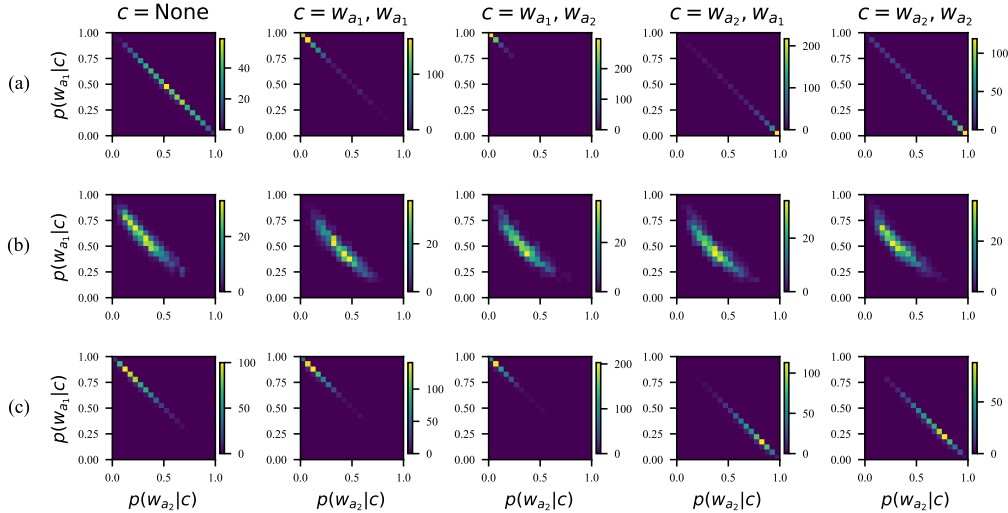

Figure 10: Initial policy distributions in the IPD for all possible two-letter action labels for (a) *gemma-2-2b-it*, (b) *Llama-3.2-1B-Instruct* with default initialization, and (c) *Llama-3.2-1B-Instruct* after SFT to match *gemma-2-2b-it*'s initialization. Each column corresponds to one of the five possible states encountered during IPD gameplay. For each state, we extract the model's initial cooperation and defection probabilities across all possible two capital letter action label pairs. Each heatmap shows the density of resulting policies, with color intensity indicating the number of action label pairs yielding each initial policy. The diagonal represents valid policies where $p(w_{a_1}|c) + p(w_{a_2}|c) \approx 1$ (minimal illegal actions).

responds to *gemma-2-2b*'s output distribution over action tokens.[13]. We fine-tune *Llama-3.2-1B-Instruct* on this dataset using a KL divergence loss that minimizes the difference between the model's output distribution and the target distribution. The action labels used in the dataset $(w_{a_1} = \text{C}, w_{a_2} = \text{D})$ are identical to the ones used for the baseline and shaping experiments.

Figure 10(c) shows the initial policies of *Llama-3.2-1B-Instruct* after SFT[14]. Despite being trained only on examples with action labels $w_{a_1} = \text{C}, w_{a_2} = \text{D}$, the fine-tuned model generalizes across action label pairs, producing distributional signatures similar to *gemma-2-2b-it*. The initial policies now concentrate near the diagonal (minimal illegal actions) and exhibit state-dependent distributions. The one exception is the stateless condition, which shows a stronger bias toward higher cooperation probabilities.

We conduct baseline and shaper experiments as per Section 5.1, with the only difference being the adapter initialization (SFT weights for *Llama-3.2-1B-Instruct* vs. random initialization for *gemma-2-2b-it*). The training prompts maintain the same structure with two model-specific modifications. First, we use the appropriate chat formatting tokens. Second, we provide the game rules and payoff matrix as a system prompt instead of a user message (since *gemma-2-2b-it* does not support system prompts, all game relevant information was passed as a user message). Figure 11 shows the initial training prompt for *Llama-3.2-1B-Instruct*. The dynamic updates of this prompt throughout training remain unchanged with respect to those used in the main experiments (see Appendix A.11).

Table 17 presents the average reward per step for both players in the baseline (two naive learners) and shaper (one shaper vs. naive learner) conditions. In the baseline, both agents converge to mutual defection with rewards of approximately 1 each, matching the *gemma-2-2b-it* baseline re-

---

[13]We extract *gemma-2-2b-it*'s probabilities for generating each of the action labels $(w_{a_1} = \text{C}, w_{a_2} = \text{D})$, renormalize them, and construct a sparse target distribution with all other vocabulary tokens receiving probability 0.

[14]SFT was performed using QLoRA with the same adapter configuration (rank-2) used in subsequent baseline and shaper experiments.

**Initial prompt for *Llama-3.2-1B-Instruct* for the IPD with $w_{a_1} = \mathbf{C}, w_{a_2} = \mathbf{D}$**

```
<|begin_of_text|><|start_header_id|>system<|end_header_id|>

You are playing a 2-player game with actions:  C, D. Points are assigned as
follows:  C/C: 3/3, C/D: 0/4, D/C: 4/0, D/D: 1/1.  Please only use C and D
to reply.<|eot_id|>.<|start_header_id|>user<|end_header_id|>

Choose an action for the current round.  Reply only with C or
D.<|eot_id|><|start_header_id|>assistant<|end_header_id|>
```

Figure 11: Initial *Llama-3.2-1B-Instruct* prompt for the IPD. The prompt is identical to that of *gemma-2-2b-it* except for the formatting tags, and the fact that the game description is given as a system prompt.

Table 17: Post-training evaluation results for the IPD comparing baseline versus shaper-naive learner pairs using *Llama-3.2-1B-Instruct* as the base model. Average rewards per step are reported with 95% confidence intervals across 5 and 10 random seeds for shaping and baseline experiments, respectively. Transitions with $a_{\text{null}}$ are excluded from the analysis ($\sim 0.1\%$ and $\sim 0.0\%$ of actions respectively).

|  | Baseline | | One Shaper | |
|---|---|---|---|---|
|  | Player 1 | Player 2 | Shaper | Opponent |
| IPD | $1.04 \pm 0.05$ | $0.99 \pm 0.02$ | $3.97 \pm 0.01$ | $0.10 \pm 0.2$ |

sults. With a shaper present, the shaper achieves 3.97, while limiting the opponent to 0.10. These outcomes closely replicate those obtained with *gemma-2-2b-it*, providing preliminary evidence of ShapeLLM's generalization across model architectures. Training dynamics are shown in Figure 12.

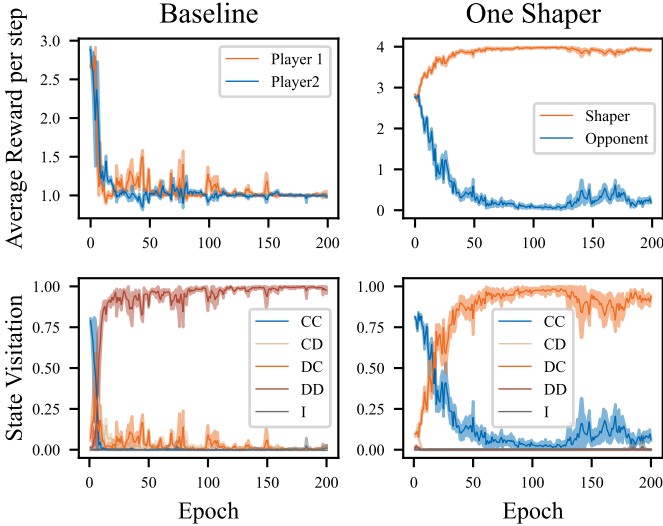

Figure 12: Average reward per step (top row) and state visitation (bottom row) during training for the baseline and shaper experiments in the IPD with *Llama-3.2-1B-Instruct* as a base model. In the state visitation figures, the outcome "I" encompasses all transitions in which either player chose $a_{\text{null}}$. The results are reported along with a 95% confidence interval over 5 random seeds.

For reproducibility, we report the hyperparameters used in these experiments. Naive learner hyperparameters follow Table 7 except for the value function coefficient, which was set to $c_{\mathrm{VF}} = 10^{-2}$ reduce instability. This modification applies to both baseline agents and to the opponent in the shaping experiments. Shaper hyperparameters follow Table 8, with $\mathrm{lr} = 3.41 \times 10^{-7}$, $c_{\mathrm{VF}} = 10^{-4}$, and $\epsilon_{\mathrm{p}} = 0.1$.

## A.10 TRAINING DYNAMICS

In this section we present the average reward per step and state visitation throughout training for various experiments discussed in the main text. These figures complement the evaluation results by showing how agent behaviors evolved during the learning process. Figure 13 presents the training dynamics for the baseline experiments across the IPD, IMP and ICG (Section 5.1). Figure 14 shows the baseline training dynamics for the cooperative shaping experiments (Section 6). Finally, Figures 15 and 16 present the training dynamics for shaper experiments against three different opponent types across the IPD, IMP, and ICG, with evaluation results reported in Section 5.2.

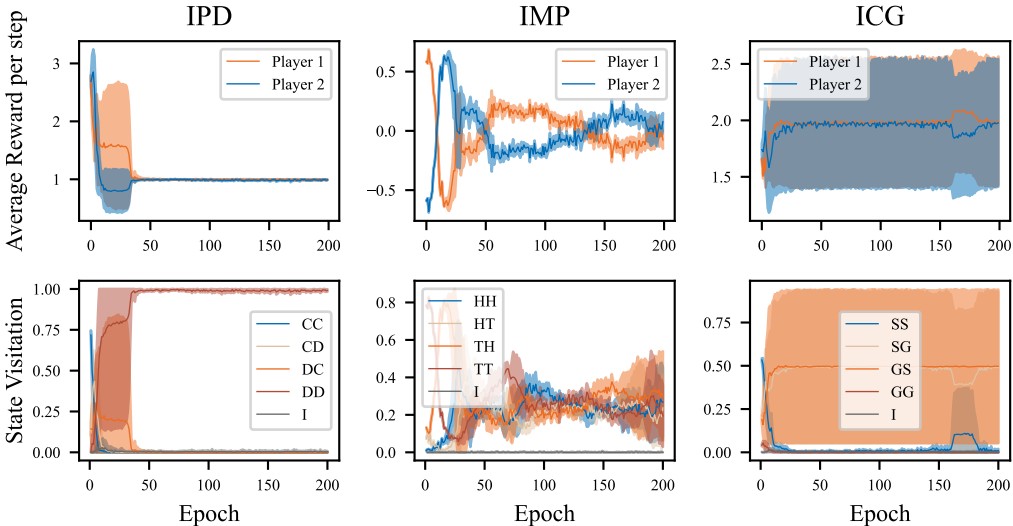

Figure 13: Average reward per step (top row) and state visitation (bottom row) during training for the baseline experiments across the IPD, IMP and ICG reported in Section 5.1. In the state visitation figures, the outcome "I" encompasses all transitions where either player chose $a_{\mathrm{null}}$. The results are reported along with a 95% confidence interval over 5 random seeds, except for the ICG experiment, for which we use 10 seeds.

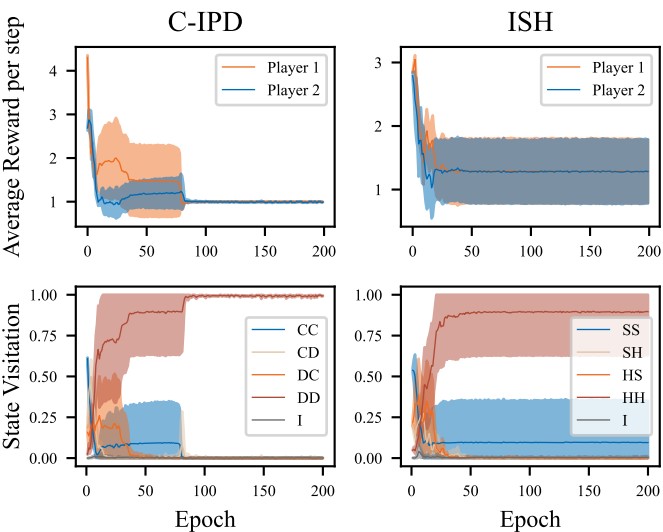

Figure 14: Average reward per step (top row) and state visitation (bottom row) during training for the baseline experiments across the C-IPD and ISH reported in Section 6. In the state visitation figures, the outcome "I" encompasses all transitions where either player chose $a_{\text{null}}$. The results are reported along with a 95% confidence interval over 10 random seeds.

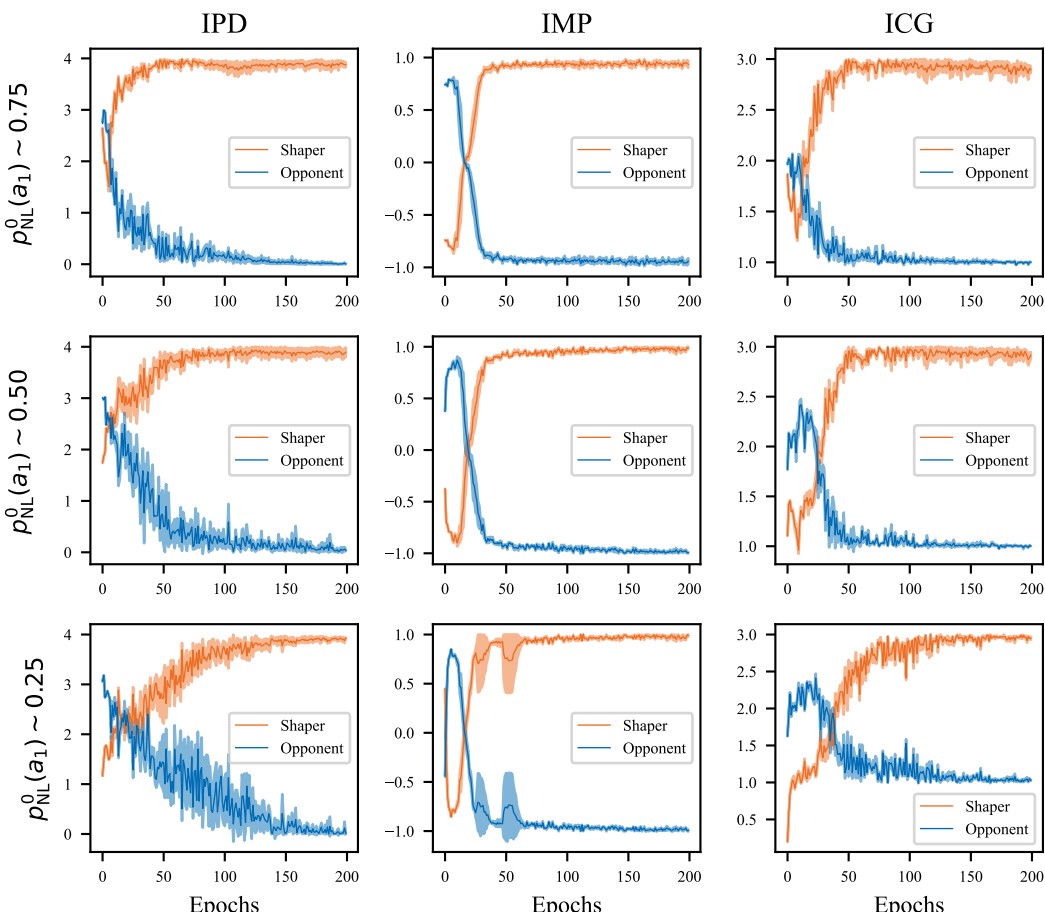

Figure 15: Average reward per step during training for the shaping experiments across the IPD, IMP, and ICG with different opponent initializations (Section 5.2). Each row corresponds to an opponent with a different initial probability of generating action $a_1$ ("Cooperate" in the IPD, "Heads" in the IMP, and "Swerve" in the ICG). The results are reported along with a 95% confidence interval over 5 random seeds.

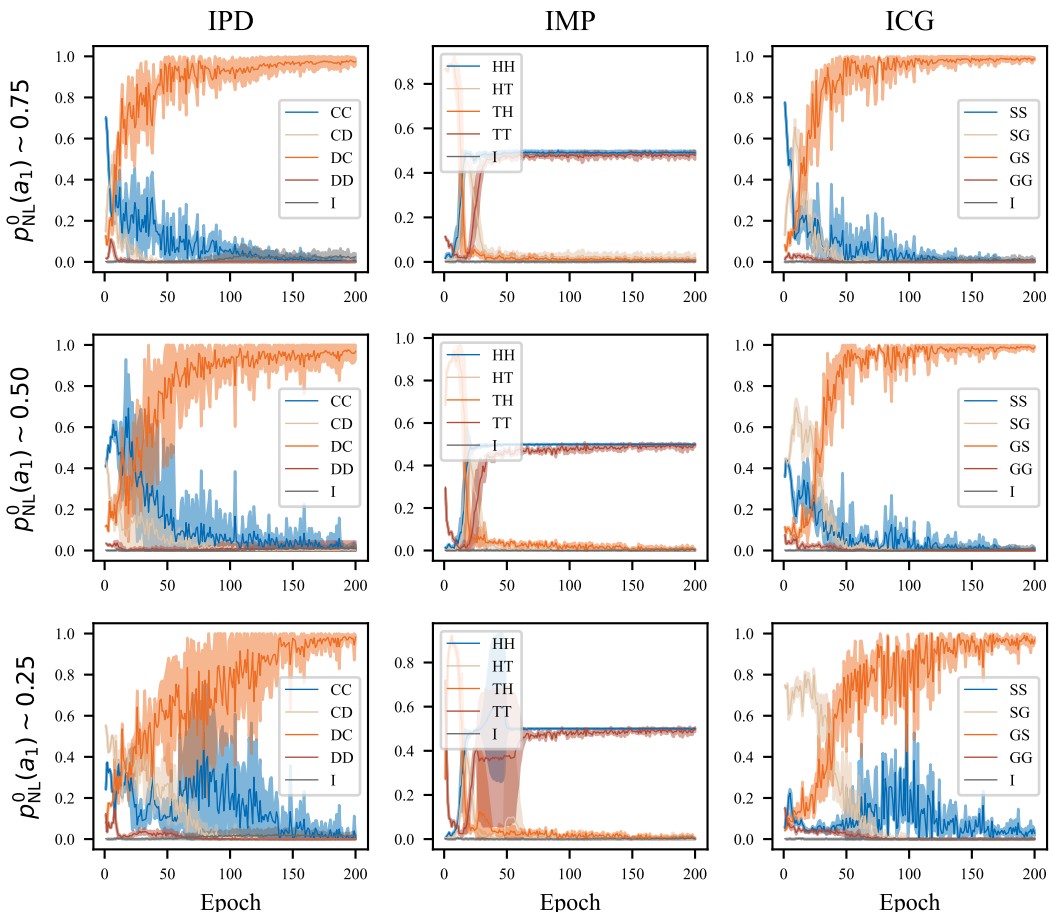

Figure 16: State visitation during training for the shaping experiments across the IPD, IMP, and ICG with different opponent initializations (Section 5.2). Each row corresponds to an opponent with a different initial probability of generating action $a_1$ ("Cooperate" in the IPD, "Heads" in the IMP, and "Swerve" in the ICG). For all games, the outcome "I" encompasses all transitions where either player chose $a_{\text{null}}$. The results are reported along with a 95% confidence interval over 5 random seeds.

## A.11 TRAINING PROMPTS

We use three types of prompts throughout training: the base prompt, the state-only prompt, and the state occurrence prompt.

The *base prompt* (Figure 17) includes a game description specifying the number of players, available actions, and payoff structure. It also contains an instruction to select an action using the designated action labels $w_{a_1}, w_{a_2}$.

The *state-only prompt* (Figure 18) includes all information from the base prompt and additionally incorporates the last joint action of both players. Naive learners receive this prompt for all game rounds except the first, where they receive the base prompt. Shapers receive this prompt only at the second round of each trial, when one previous action has been played.

Lastly, the *state occurrence prompt* (Figure 19) includes all information from the state-only prompt and additionally incorporates the state counts for all previous rounds within the current trial. Shaping agents receive this prompt for all interactions beyond the first two rounds. The prompt is dynamically updated as follows: when new actions are played, the state count is updated to include the most recent state, and the new joint action is displayed in the current state field.

---

**Base prompt for the IPD with $w_{a_1} = \mathbf{C}, w_{a_2} = \mathbf{D}$**

```
<bos><start_of_turn>user
You are playing a 2-player game with actions:  C, D. Points are assigned as
follows:  C/C: 3/3, C/D: 0/4, D/C: 4/0, D/D: 1/1.
Choose an action for the current round.  Reply only with C or D.
<end_of_turn>
<start_of_turn>model
```

---

**Base prompt for the IMP with $w_{a_1} = \mathbf{H}, w_{a_2} = \mathbf{T}$**

```
<bos><start_of_turn>user
You are playing a 2-player game with actions:  H, T. Points are assigned as
follows:  H/H: 1/-1, H/T: -1/1, T/H: -1/1, T/T: 1/-1.
Choose an action for the current round.  Reply only with H or T.
<end_of_turn>
<start_of_turn>model
```

---

**Base prompt for the ICG with $w_{a_1} = \mathbf{S}, w_{a_2} = \mathbf{G}$**

```
<bos><start_of_turn>user
You are playing a 2-player game with actions:  S, G. Points are assigned as
follows:  S/S: 2/2, S/G: 1/3, G/S: 3/1, G/G: -5/-5.
Choose an action for the current round.  Reply only with S or G.
<end_of_turn>
<start_of_turn>model
```

---

**Base prompt for the ISH with $w_{a_1} = \mathbf{S}, w_{a_2} = \mathbf{H}$**

```
<bos><start_of_turn>user
You are playing a 2-player game with actions:  S, H. Points are assigned as
follows:  S/S: 4/4, S/H: 0/3, H/S: 3/0, H/H: 1/1.
Choose an action for the current round.  Reply only with S or H.
<end_of_turn>
<start_of_turn>model
```

---

Figure 17: Base prompts for the IPD, IMP, ICG and ISH. The structure remains the same across games, with the only differences being the action labels and reward matrices.

**Example of *state-only prompt* for the IPD with $w_{a_1} = \mathbf{C}, w_{a_2} = \mathbf{D}$**

```
<bos><start_of_turn>user
You are playing a 2-player game with actions: C, D. Points are assigned as
follows: C/C: 3/3, C/D: 0/4, D/C: 4/0, D/D: 1/1.
<STATE>In the previous round, you played C and your opponent played C.
Choose an action for the current round. Reply only with C or D.
<end_of_turn>
<start_of_turn>model
```

**Example of *state-only prompt* for the IMP with $w_{a_1} = \mathbf{H}, w_{a_2} = \mathbf{T}$**

```
<bos><start_of_turn>user
You are playing a 2-player game with actions: H, T. Points are assigned as
follows: H/H: 1/-1, H/T: -1/1, T/H: -1/1, T/T: 1/-1.
<STATE>In the previous round, you played H and your opponent played H.
Choose an action for the current round. Reply only with H or T.
<end_of_turn>
<start_of_turn>model
```

**Example of *state-only prompt* for the ICG with $w_{a_1} = \mathbf{S}, w_{a_2} = \mathbf{G}$**

```
<bos><start_of_turn>user
You are playing a 2-player game with actions: S, G. Points are assigned as
follows: S/S: 2/2, S/G: 1/3, G/S: 3/1, G/G: -5/-5.
<STATE>In the previous round, you played S and your opponent played S.
Choose an action for the current round. Reply only with S or G.
<end_of_turn>
<start_of_turn>model
```

**Example of *state-only prompt* for the ISH with $w_{a_1} = \mathbf{S}, w_{a_2} = \mathbf{G}$**

```
<bos><start_of_turn>user
You are playing a 2-player game with actions: S, H. Points are assigned as
follows: S/S: 4/4, S/H: 0/3, H/S: 3/0, H/H: 1/1.
<STATE>In the previous round, you played S and your opponent played S.
Choose an action for the current round. Reply only with S or H.
<end_of_turn>
<start_of_turn>model
```

Figure 18: Example state-only prompts for the IPD, IMP, ICG and ISH. The structure remains the same across games, with the only differences being the action labels and reward matrices. The specific examples shown are for rounds in which the previous joint action is $(a_1, a_1)$.

---

**Example of *state occurrence prompt* for the IPD with $w_{a_1} = \mathbf{C}, w_{a_2} = \mathbf{D}$**

```
<bos><start_of_turn>user
You are playing a 2-player game with actions:  C, D. Points are assigned as
follows:  C/C: 3/3, C/D: 0/4, D/C: 4/0, D/D: 1/1.
<ADDITIONAL INFORMATION>The occurrence of each state in the current game has
been CC:0, CD:0, DC:0, DD:5.
<STATE>In the previous round, you played C and your opponent played C.
Choose an action for the current round.  Reply only with C or D.
<end_of_turn>
<start_of_turn>model
```

---

**Example of *state occurrence prompt* for the IMP with $w_{a_1} = \mathbf{H}, w_{a_2} = \mathbf{T}$**

```
<bos><start_of_turn>user
You are playing a 2-player game with actions:  H, T. Points are assigned as
follows:  H/H: 1/-1, H/T: -1/1, T/H: -1/1, T/T: 1/-1.
<ADDITIONAL INFORMATION>The occurrence of each state in the current game has
been HH:0, HT:0, TH:0, TT:5.
<STATE>In the previous round, you played H and your opponent played H.
Choose an action for the current round.  Reply only with H or T.
<end_of_turn>
<start_of_turn>model
```

---

**Example of *state occurrence prompt* for the ICG with $w_{a_1} = \mathbf{S}, w_{a_2} = \mathbf{G}$**

```
<bos><start_of_turn>user
You are playing a 2-player game with actions:  S, G. Points are assigned as
follows:  S/S: 2/2, S/G: 1/3, G/S: 3/1, G/G: -5/-5.
<ADDITIONAL INFORMATION>The occurrence of each state in the current game has
been SS:0, SG:0, GS:0, GG:5.
<STATE>In the previous round, you played S and your opponent played S.
Choose an action for the current round.  Reply only with S or G.
<end_of_turn>
<start_of_turn>model
```

---

**Example of *state occurrence prompt* for the ISH with $w_{a_1} = \mathbf{S}, w_{a_2} = \mathbf{H}$**

```
<bos><start_of_turn>user
You are playing a 2-player game with actions:  S, H. Points are assigned as
follows:  S/S: 4/4, S/H: 0/3, H/S: 3/0, H/H: 1/1.
<ADDITIONAL INFORMATION>The occurrence of each state in the current game has
been SS:0, SH:0, HS:0, HH:5.
<STATE>In the previous round, you played S and your opponent played S.
Choose an action for the current round.  Reply only with S or H.
<end_of_turn>
<start_of_turn>model
```

Figure 19: Example state-occurrence prompts for the IPD, IMP, ICG and ISH. The structure remains the same across games, with the only differences being the action labels and reward matrices. The specific examples shown are for rounds in which all previous joint actions within the game are $(a_2, a_2)$, except the last one, which is $(a_1, a_1)$.

