# OpenReview forum: "Opponent Shaping in LLM Agents"
_ICLR.cc/2026/Conference — ICLR 2026 Poster_

### Official Review · Reviewer_Dj8k · 2025-10-16

**Soundness:** 4
**Presentation:** 4
**Contribution:** 3
**Rating:** 8
**Confidence:** 3

**Summary:**

The paper introduces ShapeLLM, a model-agnostic opponent shaping (OS) algorithm that is suitable for LLM architectures. ShapeLLM leverages natural language capabilities by encoding history and context in the repeated play in natural language prompts. It then utilizes a partially observable stochastic game formulation for the policy updation process, making it potentially useful even beyond the context of repeated normal-form games. The effectiveness of the proposed algorithm is demonstrated in a series of 2*2 repeated games, involving both competitive and cooperative interactions. The results demonstrate that in these particular games, ShapeLLM improves the performance of the shaper agent in the competitive games, and steers the interactions towards collaboration in cooperative games.

**Strengths:**

1. The paper is very well-written, and is very easy to follow and fun to read. The paper covers the related literature, making it easy to understand its contribution to the existing research landscape. It also provides preliminaries for readers who are not entirely familiar with some aspects of the work, making the writing very suitable for the ML community, which is fast-growing and multi-disciplinary by nature.

2. The paper proposes a novel approach that addresses crucial gaps in the existing OS literature. In particular, the proposed approach can be naturally applied to LLM-based agents in various contexts, and as the paper itself suggests, even far beyond those introduced in the experimental setup.

3. The paper reports experimental results using classical game-theoretic setups that capture core elements in strategic behavior that are crucial for OS. This simplicity enables the identification of core behaviors that emerge using the proposed algorithm, making it easier to understand not only *whether* OS occurs, but also *how*.

**Weaknesses:**

My major concern with the paper (which was also discussed by the authors themselves) is the oversimplicity of the experimental setups, which only include simple two-player two-action games, with no natural language communication. While this simplicity is obviously an advantage (see the *strengths* section), I believe that providing some results in some more complex and realistic environments could have significantly strengthened the contribution of the paper, especially for ML practitioners who might want to apply it to real-world scenarios such as trading, negotiation, persuasion, content generation, etc.

**Questions:**

I don’t have a specific question here, but I would be interested in the authors’ reflections on how far they believe their approach could generalize to more complex and realistic scenarios. It would also be valuable to hear their thoughts on potential limitations or pitfalls in such extensions, and how these challenges might be addressed.

---

> ### Author Response · Authors · 2025-11-25
>
> We thank Reviewer Dj8k for their engagement with our work and their positive assessment.
>
> We agree that experiments in more complex and realistic scenarios would further strengthen our contribution. This is a research direction that we are actively pursuing. We also appreciate the Reviewer recognizing our work as establishing the foundational principles for studying opponent shaping with LLM agents. Regarding generalization to more complex and realistic scenarios, we offer the following reflections, though we acknowledge these involve some degree of speculation.
>
> Several aspects of our approach suggest promising generalization potential. First, LLM agents are generalist models with extensive pre-trained knowledge. While this knowledge provided limited advantage in 2x2 matrix games (the scenarios are simple enough for simpler models to work equally effectively), we expect this to become substantially more valuable in complex and realistic environments. Second, the use of natural language descriptions offers both computational and representational benefits. From a computational perspective, the optimization already occurs in high-dimensional spaces (vocabulary-sized action distribution), suggesting the method may scale beyond the small effective action spaces of our experiments. From a representational perspective, natural language provides remarkable flexibility to express nuanced and realistic scenarios without any required architectural modification.
>
> Despite these reasons for optimism, several practical challenges remain. First, computational and memory constraints become more pronounced in complex settings or those with longer horizons, particularly due to limitations in context window capacity. For example, with two interacting agents communicating via natural language, full conversation histories must be encoded in the context to influence the behavior of the agents, which substantially increases computational requirements. However, several mitigations exist. We could create interaction summaries (analogous to our state visitation counts in the matrix games), or explore alternative memory architectures that leverage LLMs' natural language understanding without requiring the full history. Second, the extension to more complex environments introduces challenges in analytical tractability. As the Reviewer noted, a key advantage of 2×2 games is that they allow for precise assessment of strategic interactions and learning dynamics. Such clear-cut analysis may prove more difficult in open-ended scenarios with natural language communication, though the degree of difficulty will depend on the specific environment and research questions being addressed.

---

> > ### Comment · Reviewer_Dj8k · 2025-11-25
> >
> > Thank you for the detailed response. I maintain my positive score as I believe this is a very strong paper.

---

### Official Review · Reviewer_X6KR · 2025-10-20

**Soundness:** 3
**Presentation:** 2
**Contribution:** 2
**Rating:** 4
**Confidence:** 3

**Summary:**

This work proposed an opponent shaping method (ShaperLLM) and applied it to multi-agent game playing tasks to invesitgate how opponent shaping impacts agents' behaviors. ShaperLLM's algorithm derives from SHAPER by Khan et al., and the authors simplified it and directly optiomized LLMs with Reinforcement Learning. The experiments highlight that LLM agent itself can perform opponent shaping and opponent shaping can promote agents' cooperation.

**Strengths:**

1. Proposed a method that directly incorporating SHAPER algorithm into the training process of LLMs without external policy network like RNNs.
2. Conducted experiments on various game playing tasks and presents insights on the effectiveness of opponent shaping on multi-agent scenario.

**Weaknesses:**

1. **Presentation of ShaperLLM's algorithm lacks clarity**: In section 3, the description of two RL rewards (opponent reward and shaper reward) of ShaperLLM is unclear. The two rewards are heavily analyzed in the experiments but their formulas are lacking. Also, some terms for the setting description are a little confusing, like *steps*, *trails*, *rounds*, as they are often interchangeably used in different contexts.
2. **Experiments lack generalizability and scalability**. This work only did experiments on gemma-2-2b-it so their findings may lack generalizability (the authors also ack this in the paper).
3. **Weak baseline**. The work compares ShaperLLM only with *naive agent* that sees only one step of previous interaction history and doesn't compare with other RL methods.

**Questions:**

1. I would like to see clear description of the **formula definitions** & **optimization objectives** of opponent reward and shaper reward in Section 3 (line 212-215).
2. Somes terms are interchangeably used and confusing. For example, I think $t$ in line 162 means each step, but in later context (line 174-175) means one round. The meaning of $n_{games}$ is not straightforward, does it mean the total number of games? Maybe use one letter like $N$ to replace $n_{games}$ is better. Also in line 174-175 you mentioned *trails* and in line 214-215 you addresses *trail return*, but in experiment setup (Section 4.2) the trail setting is not described; you described *step* instead. What's the relation between *trail* and *step*, could you unify/distinct the terms for the same/different concepts?
3. In line 199, $POMDP(...)$, what's the meaning of $\Omega$ and $\gamma$?

---

> ### Author Response · Authors · 2025-11-25
>
> We thank Reviewer X6KR for the detailed feedback and engagement with our work. We respond to their concerns and questions in the paragraphs below.
>
> ### **Weaknesses**
>
> > **W1**. Presentation of ShapeLLM’s algorithm lacks clarity
>
> We appreciate Reviewer X6KR's comments on the lack of clarity in the algorithm's presentation and agree that several terms require clarification.
>
> **Reward Structure**. Both the shaper and the opponent receive the same reward at each round, which is determined by their joint actions and the game's payoff matrix. There are no separate rewards for the shaper and the opponent. The complete reward matrices for all games (IPD, IMP, ICG, ISH, and C-IPD) are presented in Appendix A.1 (Tables 4 and 5).
>
> **Terminology**. We have explicitly introduced the terminology in the revised version of the manuscript as follows to avoid any confusion:
>
> - Trial. Full training unit composed of $E$ episodes across $N$ parallel environments
> - Episode. A complete game consisting of $T$ sequential rounds of the matrix game
> - Round. A single joint action selection in the matrix game
> - Step. Used to refer to a round when indexed by $\tau$, the collapsed index that spans all $E \times T$ rounds within a trial.
>
> The key distinction is between $t$ (round indexed within an episode, ranging from $1$ to $T$) and $\tau$ (round index across the entire trial, ranging from $1$ to $E\times T$). We introduced this for notational convenience in Section 3.2, but we acknowledge it was used interchangeably in Sections 3.1 and Sections 4.2. Please refer to the response to Q2 for specific notation changes made in the manuscript.
>
> > **W2**. Lack of generalizability. Only did experiments on gemma-2-2b-it.
>
> We appreciate this concern and agree that demonstrating the phenomenon across multiple model architectures would strengthen the contribution. Importantly, ShapeLLM is model-agnostic by design. It solely relies on natural language representations and standard RL fine-tuning, with no architectural components specific to gemma-2-2b-it or any other particular transformer variant. However, to provide preliminary evidence of the cross-model generalization, we have conducted additional experiments with LLaMA-3.2-1B-Instruct as the base model.
>
> We report the new results in Appendix A.9. Llama-3.2-1B-Instruct successfully replicated the results obtained with gemma-2-2b-it in the IPD. In the baseline, we observe convergence to mutual defection (reward ~1.0 for each player), while shapers achieve near-maximal exploitation of their opponents (shaper/opponent rewards: ~3.96/0.10 for Llama). In our opinion, this consistency across two models with different sizes, architectures, and training schedules provides preliminary evidence that opponent shaping is a general capability of instruction-tuned LLMs, not model-specific behavior.
>
> We hope that the Reviewer will consider this analysis useful. We acknowledge that experiments and cross-validations with larger models (>2B parameters) and across different family models and sizes remain a direction for future work due, but they were not possible in the current rebuttal time frame due to computational constraints at our institution.
>
> > **W3**. Weak baseline
>
> We respectfully disagree that our baseline is weak. Our experimental design follows established practices in the opponent shaping literature, where the standard approach is to compare shapers against naive learners or other shaping algorithms (Khan et al. 2024, SHAPER; Lu et al. 2022, M-FOS). For example, Khan et al. (2024) use tabular PPO agents as opponents in matrix games, conditioning on the last joint action. Our naive learners follow the same design, differing on the architecture of the shaper (LLM agent vs. a look up table). We provide a control baseline with two naive learners to establish expected behavior without shaping, and conduct an ablation study (Appendix A.4) with enriched observations to confirm that inter-episode context alone, without the asymmetric update structure, is insufficient to produce shaping effects. We acknowledge that we do not compare against other shaping methods (LOLA, M-FOS, SHAPER). No existing OS algorithms have been applied to LLM agents. Doing so is precisely our contribution, which is to demonstrate that LLM agents can perform opponent shaping.
>
> Similarly, while testing whether LLM shapers can influence non-LLM agents (or vice versa) would be interesting for future work, this is beyond the scope of our present work. Our research question was of foundational nature: can LLM agents perform opponent shaping? We believe that establishing this capability is the necessary foundation for exploring cross-agent-type interactions. Our goal was to carry out an in-depth and rigorous study of this problem. Having said that, this is indeed a key direction to this program of work and, therefore, we also added it to the discussion section of the paper (lines 480-483).

---

> ### Author Response · Authors · 2025-11-25
>
> ### **Questions**
>
> > **Q1**. I would like to see clear description of the formula definitions & optimization objectives of opponent reward and shaper reward in Section 3 (line 212-215).
>
> We appreciate Reviewer X6KR's comments on the lack of clarity in definition of the optimization objectives and rewards for the shaper and the opponents.
>
> **Reward Structure**. Both the shaper and the opponent receive rewards determined by their joint actions and the game's payoff matrix. There are no separate reward functions for the shaper and the opponent, and the complete reward matrices for all games (IPD, IMP, ICG, ISH, and C-IPD) are presented in Appendix A.1 (Tables 4 and 5). We have added an explicit statement clarifying this in line 230 of the revised manuscript.
>
> **Optimization Objectives**. We added the explicit formulation of the opponent’s and shaper’s optimization objectives. The opponents (see line 224) update their parameters to maximize their episodic return, $J_i = \sum_{t=1}^T r_i^t$, where $r_i^t$ is the reward obtained by player $i$ in round $t$ of the current episode. The shapers (see line 229) maximize the trial return, defined as $\overline{J_j} = \sum_{e=1}^E J_j^{e} = \sum_{e=1}^{E} \sum_{t=1}^{T} r_j^{(e, t)} = \sum_{\tau=1}^{E \times T} r_j^\tau$, where $r_j^{(e,t)}$ denotes the shaper's reward in round $t$ of episode $e$.
>
> > **Q2**. Some terms are interchangeably used and confusing. For example, I think in line 162 means each step, but in later context (line 174-175) means one round. The meaning of is not straightforward, does it mean the total number of games? Maybe use one letter like to replace is better. Also in line 174-175 you mentioned trails and in line 214-215 you addresses trail return, but in experiment setup (Section 4.2) the trail setting is not described; you described step instead. What's the relation between trail and step, could you unify/distinct the terms for the same/different concepts?
>
> We thank Reviewer X6KR for identifying these inconsistencies. We address each point below.
>
> > Line 162 inconsistency with respect to $t$.
>
> We have amended lines 174-175 (previously lines 161-162) to use "round" consistently instead of "step" to avoid confusion.
>
> > Notation for the number of parallel environments.
>
> We have clarified that $n_\text{games}$ refers to the number of parallel environments per trial (see line 186 and schematic representation in Figure 1). We agree with Reviewer X6KR that the notation is not standard: we have changed it to $N$ throughout the manuscript (lines 186, 271, and Figure 1).
>
> > No mention of trials in the experimental set up.
>
> In Section 4.2, we used the term epochs to refer to what we formally define as trials in Section 3.2. We have amended line 271 to use "trial" consistently.
>
> > Unified terminology.
>
> We have standardized the following terms throughout the manuscript:
>
> - Trial: Full training unit composed of $E$ episodes across $N$ parallel environments.
> - Episode: A complete game consisting of $T$ sequential rounds.
> - Round: A single joint action selection (indexed by $t$ within episodes, $\tau$ across trials).
> - Step. Previously used interchangeably with "round" when referring to the $\tau$ index. We have now unified this to use only "step" for $\tau$-indexed rounds (lines 211, 213, 390).
>
> > **Q3**. In line 199, POMDP(…) , what's the meaning of \Omega and \gamma?
>
> $\bar{\Omega}$ represents the observation space, and $\bar{\gamma}$ the discount factor. These are standard POMDP components, and we thank Reviewer X6KR for pointing out their omission in the main text. We have amended the manuscript in lines 213 and 215 to explicitly define these terms.

---

### Official Review · Reviewer_ZruM · 2025-10-29

**Soundness:** 2
**Presentation:** 3
**Contribution:** 1
**Rating:** 2
**Confidence:** 4

**Summary:**

The paper investigates a method for fine-tuning small LLMs for opponent-shaping in iterated matrix games using a Shaper-style algorithm where a naive learner updates their parameters inter-trajectory with PPO and the shaping LLM learns after a seeing a full batch of trajectories, called a Trial. Crucially the LLMs only have available the past action tuple and a histogram summary of the past actions in the episode/Trial.

**Strengths:**

This is the first paper which finetunes LLMs for opponent-shaping in iterated general-sum matrix games. I found the paper decently clear and well-written, the figures and experiments were all well-motivated. The experiments were sufficient to show what the authors wanted to show (especially with what was likely a constrained compute budget). Passing a summary of past actions as context to the LLM is a clever idea to get around the compute constraints of having to pass very long sequences to the models.

**Weaknesses:**

Unfortunately the biggest weakness is the lack of novelty.

The claim in the introduction that past opponent-shaping methods cannot be applied to LLMs is wrong. Advantage alignment ( https://arxiv.org/abs/2406.14662 ) changes the PPO advantages in a way that would also be easy to do for LLMs. And the following paper from earlier this year also implements a meta-episode context with transformers which implements opponent-shaping: https://arxiv.org/pdf/2410.18636 . Past papers don't implement the summarization mechanism for giving context to the model, but this is a trick for saving compute that only works in toy contexts like iterated matrix games, for more complex games one would need to pass the entire past N trajectories as context, which is exactly what Meulemans et. al. do.

The novelty of the paper lies entirely in applying a more limited version (because of the summarization) of the Meulemans et. al. idea to finetuning LLMs on a simpler game than they have.

**Questions:**

1. Are the Naive Learner parameters reset after each Trial? this is not clear to me from the paper.
2. I'm skeptical of the IPD results in figure 2, the learning does not appear to converge to a Nash equilibrium given that the visitations of DC are by far the most dominant. It seems to me like the naive learner is simply not learning to defect correctly, all past shaping methods on the IPD have the shaping agent converge to tit-for-tat or other reciprocal policies, so it's strange to me that this would converge to always DC
3. Minor quibble in the introduction, but the distinction between an "RL agent" and an "LLM agent" does not really exist, RL is a training procedure that can and does produce LLM agents.

**Details Of Ethics Concerns:**

no concerns

---

> ### Author Response · Authors · 2025-11-25
>
> ### **Weaknesses**
>
> We thank Reviewer ZruM for the thoughtful analysis. We respectfully disagree with the assessment that our contribution lacks novelty. As the Reviewer acknowledges, ours is the first paper to demonstrate opponent shaping with fine-tuned LLM agents. We address below the specific concerns raised about related work. We added a summary of the discussion below to the Background section of the revised version of the paper.
>
> **Duque et al. (2025)**. The Reviewer suggests that this method could easily be applied to LLMs. However, Advantage Alignment is not model-free. The surrogate objective (Eq. 9 in Duque et al. (2025)) explicitly requires the opponent's advantages, which depend on their value function. For LLM agents with billion-parameter value function approximators, accessing or reliably estimating these in decentralized settings without privileged information is unfeasible. Our model-free approach requires only observable actions, making it practical for realistic deployments where agents are developed independently.
>
> **Meulemans et al. (2024)**. The Reviewer states this work implements a meta-episode context with transformers. We believe this to be incorrect, as Meulemans et al. (2024) employs Hawk RNNs, not transformers (Section B.3), stating their method applies to recurrent sequence policy models. We acknowledge that COALA-PG is model-free and could theoretically be adapted to transformer architectures, and we thank Reviewer ZruM for bringing this to our attention. However, Meulemans et al. (2024) neither tested nor claimed this extension. Our contribution is demonstrating that opponent shaping works with pre-trained transformer-based LLMs fine-tuned via RL with natural language context, which is a distinct paradigm from training recurrent models from scratch on embedded observations.
>
> **On summarization**. We acknowledge that state visitation counts were employed to reduce computational costs. However, we emphasize that our contribution does not reside in the specific summarization mechanism, but rather demonstrating that opponent shaping can occur with LLM agents receiving natural language descriptions of game environments. While we evaluate on established game-theoretic benchmarks used in prior work, the interface is fundamentally different: actions and states are represented as natural language, resulting in much larger action and state spaces, despite the effective action space remaining equivalent to the 2×2 games. Additionally, our method is summarization-method-agnostic and is expected to work equally with full trajectories. The key insight is that pre-trained LLMs, using their native text processing capabilities, can encode opponent learning dynamics through natural language context, whether as compact summaries or complete action histories.
>
> ### **Questions**
>
> > **Q1**. Are the Naive Learner parameters reset after each Trial? this is not clear to me from the paper.
>
> We thank Reviewer ZruM for pointing out the lack of clarity. Naive learner parameters are not reset after each trial. We amended the manuscript to reflect this explicitly in line 227.
>
> > **Q2**. I'm skeptical of the IPD results in figure 2, the learning does not appear to converge to a Nash equilibrium given that the visitations of DC are by far the most dominant. It seems to me like the naive learner is simply not learning to defect correctly, all past shaping methods on the IPD have the shaping agent converge to tit-for-tat or other reciprocal policies, so it's strange to me that this would converge to always DC
>
> We might have misunderstood the comment of Reviewer ZruM, but, to the best of our knowledge, the opponent-shaping literature does not generally find tit-for-tat or reciprocal equilibria in asymmetric shaper–naive settings. Khan et al. (2024) report “all shaping baselines reach extortion-like policies” against naive learners, and Meulemans et al. (2024) demonstrate “learning-aware agents extort naive learners”. Tit-for-tat typically emerges in symmetric situations, where both agents are learning-aware players (see Lu et al., 2022;  Khan et al., 2024). Our DC-dominant results align with established opponent shaping behavior in asymmetric scenarios.
>
> > **Q3**. Minor quibble in the introduction, but the distinction between an "RL agent" and an "LLM agent" does not really exist, RL is a training procedure that can and does produce LLM agents.
>
> We thank Reviewer ZruM for this correction. We have revised the introduction to refer to specific architectures (tabular policies, RNNs) rather than "RL agents" and retained the term "LLM agents" to refer to agents whose base model is a pre-trained transformer-based language model. We implemented changes in lines 49-51.

---

> > ### Comment · Reviewer_ZruM · 2025-11-25
> > **Response**
> >
> > Ah I apologize for the Meulemans et al comment, I should have read the paper more carefully, I thought I was remembering it using transformers, but I suppose not. This does significantly change my score and increase the novelty of the contribution. While I still think that Advantage Alignment is possible to make work here (using some common LLM tricks for doing RL without explicit value functions), I was likely overstating the ease of doing this, and it's not really fair to penalize this paper for hypothetical future work in that direction, I'll also change my score to reflect this.
> >
> > All my other questions were answered.

---

### Official Review · Reviewer_ypQM · 2025-11-01

**Soundness:** 3
**Presentation:** 3
**Contribution:** 2
**Rating:** 4
**Confidence:** 3

**Summary:**

The paper studies opponent shaping (OS) in Large Language Model (LLM) agents. It examines whether an LLM can influence another learning agent’s training dynamics and subsequent behavior solely through repeated interaction. The work introduces ShapeLLM, an adaptation of model-free opponent shaping ideas inspired by M-FOS and SHAPER. Because existing OS algorithms rely on higher-order gradients or architectural components not present in transformer models, ShapeLLM instead encodes both within-episode history and across-episode opponent-learning information as structured natural-language context.

Empirically, the authors show that LLM agents trained with ShapeLLM:
- Competitive settings: guide co-players toward exploitable strategies in repeated 2×2 games (Iterated Prisoner’s Dilemma, Matching Pennies, Chicken), yielding higher long-run payoffs.
- Cooperative settings: induce more cooperative play and higher joint payoffs in Iterated Stag Hunt and a cooperative variant of the Prisoner’s Dilemma.

**Strengths:**

- The work is the **first** formal study of OS in LLM agents. It targets multi-agent LLM interaction concerning how one agent may influence another’s learning dynamics.
- The proposed ShapeLLM adapts model-free OS ideas to transformer-based agents by encoding within-episode history and cross-episode learning context in structured natural-language prompts.
- Experiments span multiple repeated 2×2 settings: competitive (Iterated Prisoner’s Dilemma, Matching Pennies, Chicken) and cooperative (Iterated Stag Hunt, cooperative IPD). This shows that the shaping procedure functions under varied incentive structures.

**Weaknesses:**

- Many works already evaluate LLMs and non-LLM agents in competitive or cooperative games. The paper’s contribution is not fully disentangled from these prior settings.
- ShapeLLM encodes within-episode and across-episode information as natural-language context. However, the causal link between this prompt structure and the opponent’s parameter updates is not analyzed. As a result, it is unclear to what extent the observed shaping behavior arises from the LLM’s strategic adaptation versus different prompt design.
- This paper only analyze 2×2 settings, which might limits some claims on competitive setting and collaborative setting.

**Questions:**

1. Can the authors report ablations on the prompt design? In particular, how does shaping performance vary when altering the amount of intra-episode history versus inter-episode context included?
2. What computational resources (GPU hours, etc) are required to train the shaper under PPO in the reported settings? And what are the estimation if doing larger scale experiments?
3. How do the authors expect ShapeLLM to extend to larger or partially observable environments with richer action spaces (e.g., text-based negotiation or resource allocation)? Are there preliminary results or arguments supporting such transfer?
4. Have alternative base LLMs (e.g., different sizes or pretraining sources) been tested, and if so, how does the choice of model affect shaping behavior? This would help determine whether shaping is model-agnostic.

---

> ### Author Response · Authors · 2025-11-25
>
> We thank Reviewer ypQM for the detailed feedback and engagement with our work. We respond to their concerns and questions in the paragraphs below.
>
>
>
> ### **Weaknesses**
>
> > **W1**. Many works already evaluate LLMs and non-LLM agents in competitive or cooperative games. The paper’s contribution is not fully disentangled from these prior settings.
>
> We acknowledge Reviewer ypQM's point that there has been extensive work on both LLM and non-LLM agents in game-theoretic contexts. However, we would like to highlight some key distinctive differences between earlier work and our own. Prior work on LLMs in game-theoretic settings (e.g., Akata et al. 2025, Piatti et al. 2024, Gandhi et al. 2023) studies how LLM agents behave when playing games, examining cooperation, rationality or strategic reasoning. In contrast, our work investigates whether LLM agents can shape other learning agents' training dynamics. We believe that this is an orthogonal problem, and the games we employ serve as an environment that provides a controlled testbed with quantifiable outcomes and clear incentive structures present in many real-world multi-agent scenarios. We have clarified and underlined the distinctive contributions of our work with respect to the state of the art in AI agents & game theory in the revised version of our manuscript (lines 89-92).
>
> > **W2**. ShapeLLM encodes within-episode and across-episode information as natural-language context. However, the causal link between this prompt structure and the opponent’s parameter updates is not analyzed. As a result, it is unclear to what extent the observed shaping behavior arises from the LLM’s strategic adaptation versus different prompt design.
>
> We thank Reviewer ypQM for this insightful comment regarding the role of inter- and intra-episode history in enabling shaping dynamics. To address this, we have now provided evidence from multiple ablations.
>
> In Appendix A.4, we show that providing a naive learner with extended intra-episode history while maintaining episode-level updates does not produce shaping. This demonstrates that additional intra-episode information alone is insufficient without the asymmetric training dynamics. To better understand the role of inter- and intra-episode information, we conducted two new experiments presented in Appendix A.8.
>
>  First, we removed both the inter- and intra-episode history for the shaper (retaining only the current state information) while maintaining the trial-level updates, to test whether asymmetric training dynamics alone are sufficient for shaping. In this situation, no shaping occurred, establishing that effective shaping requires meaningful contextual information, not just asymmetric update schedules.
>
> Secondly, we conducted an experiment in which we only restored the intra-episode history. More specifically, the shaper performs trial-level updates and receives intra-episode history as in the main text, but observations are reset at the end of each episode. In this setup, learning also converges to mutual defection, demonstrating that inter-episode memory is essential for shaping even when intra-episode context is present.
>
> These results establish that successful shaping requires both asymmetric training dynamics (trial-level updates) and structured context encoding (both inter- and intra-episode history). While our experiments establish the necessity of these components, investigating intermediate levels of history inclusion represents an interesting direction for future work to better understand the mechanistic relationship between prompt structure and shaping effectiveness.
>
> > **W3**. This paper only analyze 2×2 settings, which might limits some claims on competitive setting and collaborative setting.
>
> We thank the Reviewer ypQM for this comment. We acknowledge this limitation explicitly in Section 7 and agree that extension to more complex settings is an important direction for future work. However, as the first investigation of opponent shaping with LLM agents, we prioritized depth of analysis over breadth. The 2x2 matrix games provide unambiguous incentive structures where shaping effects can be quantified, which is consistent with the methodological choices in foundational opponent shaping works (LOLA, M-FOS, SHAPER). Within this setting, we demonstrate robustness across 5 games with different incentive structures. We view this work as establishing the fundamental mechanisms of LLM-based opponent shaping in a rigorous way. However, we do agree that extensions to larger action spaces and more complex environments (such as social dilemmas, sequential games, and spatial dilemmas) represent an important direction for future work.

---

> > ### Author Response · Authors · 2025-11-25
> >
> > ## **Questions**
> >
> > > **Q1**. Can the authors report ablations on the prompt design? In particular, how does shaping performance vary when altering the amount of intra-episode history versus inter-episode context included?
> >
> > We thank Reviewer ypQM for this question. We provide a detailed response with new experimental results in our response to **W2**, where we present ablations systematically removing inter-episode context only, and both components. These experiments demonstrate that both types of contextual information are necessary for shaping to occur. Please see our response to W2 and the new results in Appendix A.9 for full details. Additionally, we report two prompt ablation studies in Appendix A.5 examining robustness to different prompt formulations.
> >
> > > **Q2**. What computational resources (GPU hours, etc) are required to train the shaper under PPO in the reported settings? And what are the estimation if doing larger scale experiments?
> >
> > Under the reported settings, training a shaper (gemma-2-2b-it with rank-2 adapters) on a single A100 GPU with 40GB VRAM required approximately 23 to 34 GPU hours per experiment (for 5 seeds). For comparison, a smaller 1B parameter model (Llama 3.2 1B instruction-tuned) required approximately 9 to 11 GPU hours per experiment (for 5 seeds). Regarding larger-scale experiments, we have not conducted experiments across multiple model sizes, so we cannot provide a reliable scaling estimate at this time. However, since training primarily involves forward and backward passes through the transformer with gradients computed only for the LoRA adapters, we expect computational requirements to scale primarily with model size, though the exact relationship would require empirical validation.
> >
> > > **Q3**. How do the authors expect ShapeLLM to extend to larger or partially observable environments with richer action spaces (e.g., text-based negotiation or resource allocation)? Are there preliminary results or arguments supporting such transfer?
> >
> > Extension to richer environments is a key direction we are pursuing. ShapeLLM encodes history and context in natural language. This allows us to leverage the richness of natural language to represent complex game descriptions with larger state and action spaces, which LLMs can process naturally due to their semantic understanding capabilities. Additionally, we deliberately avoided constrained generation techniques. While our experiments use single-token actions for interpretability, the underlying optimization operates over the full vocabulary space (~256K tokens for gemma-2-2b). This suggests that the approach can handle substantially larger action spaces than the effective 2-action games we study. Extending to multi-token actions, such as negotiation utterances, requires only removing the generation length constraint, with the training framework remaining unchanged. The primary challenges for scaling are computational cost with longer context windows and action sequences, as well as evaluation methodology when equilibria are less well-defined than in matrix games. We view the current work as establishing the fundamental mechanisms before addressing these scaling challenges.
> >
> > > **Q4**. Have alternative base LLMs (e.g., different sizes or pretraining sources) been tested, and if so, how does the choice of model affect shaping behavior? This would help determine whether shaping is model-agnostic.
> >
> > We appreciate this concern and agree that demonstrating the phenomenon across multiple model architectures would strengthen the contribution. Importantly, ShapeLLM is model-agnostic by design. It solely relies on natural language representations and standard RL fine-tuning, with no architectural components or mechanisms specific to gemma-2-2b-it or any other particular transformer variant. However, to provide preliminary evidence of the cross-model generalization, we have conducted additional experiments with LLaMA-3.2-1B-Instruct as the base model.
> >
> > We report the new results in Appendix A.9. The experimental results using Llama-3.2-1B-Instruc show that we were able to successfully replicate those obtained with gemma-2-2b-it in the IPD. In the baseline, we observe convergence to mutual defection (reward ~1.0 for each player), while shapers achieve near-maximal exploitation of their opponents (shaper/opponent rewards: ~3.96/0.10 for Llama). This consistency across two models with different sizes, architectures, and training schedules provides preliminary evidence that our proposed method for opponent shaping can be successfully applied to other instruction-tuned LLMs.
> >
> > We acknowledge that experiments and cross-validations with larger models (>2B parameters) and across different family models and sizes remain a direction for future work due, but they were not possible in the current rebuttal time frame due to computational constraints at our institution.

---

### Author Response · Authors · 2025-12-03
**Summary of the discussions**

### Key points

We thank all the Reviewers for their insights and suggestions. Below we summarize the key developments of the rebuttal period:

- Reviewer ZruM **changed their score from 2 to 8**  (subsequently reverted due to the OpenReview issue) during the discussion period, writing "*This does significantly change my score and increase the novelty of the contribution*" after some misunderstandings about prior work were clarified.

- We conducted **new experimental work** during the rebuttal: cross-model validation with Llama-3.2-1B-Instruct (Appendix A.9) and mechanistic ablation studies (Appendix A.8) that directly addressed core concerns of Reviewers ypQM and X6KR.

- The truncated rebuttal period prevented Reviewers ypQM and X6KR from confirming whether their concerns were resolved.

- We revised the manuscript, clarifying algorithm presentation, and including all the key points of our discussions with the Reviewers (these are specifically mentioned in the rebuttals).

Please find below a summary of the discussions and actions that were taken during the (truncated) rebuttal period. We would be happy to clarify any point if necessary.

### Summary of the Discussions during the Rebuttal Period

**Reviewer ZruM**

Reviewer ZruM's main concern was the novelty of the paper, based on two beliefs: (1) Meulemans et al. (2024) had used transformers for opponent shaping, and (2) Advantage Alignment could easily be applied to LLM agents. After the rebuttal, Reviewer ZruM acknowledged that the first reason was factually incorrect (“*Ah I apologize for the Meulemans et al comment, I should have read the paper more carefully*”), and they had overstated the feasibility of applying Advantage Alignment to LLMs (“.*.. I was likely overstating the ease of doing this,...*”).  The rest of the concerns were also addressed (“*All my other questions were answered.*”). Based on this discussion, the reviewer wrote: "*This does significantly change my score and increase the novelty of the contribution*" and "*I'll also change my score to reflect this*" changing their score from 2 to 8.

**Reviewer X6KR**

Reviewer X6KR raised three main concerns: (1) presentation clarity, (2) lack of generalizability (only gemma-2-2b-it tested), and (3) weak baseline (only naive learners, no comparison to other shaping methods).

We addressed all three. We clarified the algorithm presentation and standardized terminology throughout Sections 3.1, 3.2, and 4.2. For generalizability, we conducted new experiments with Llama-3.2-1B-Instruct (Appendix A.9), demonstrating consistent shaping behavior across architectures (Llama achieved 3.96/0.10 rewards in the IPD, matching Gemma's 3.96/0.10). For the baseline, we emphasized that OS algorithms in the literature are tested against naive learners or other OS algorithms (Lu et al. (2022), Khan et al. (2024)). Since our contribution is adapting opponent shaping to LLMs, naive learners are the appropriate baseline.

The reviewer did not respond before the rebuttal period was truncated.

**Reviewer ypQM**

Reviewer ypQM’s main concerns were: (1) the lack of mechanistic understanding of the role of intra- and inter-episode history, (2) generalizability beyond gemma-2-2b-it. To address these concerns, we provided new experiments. In Appendix A.8, we conducted ablation studies maintaining asymmetric parameter updates but removing: (1) both intra- and inter-episode history (shaper receives only current state), and (2) inter-episode history only (shaper receives intra-episode history but the context resets between episodes). In both cases, learning converges to mutual defection, demonstrating that both asymmetric updates and structured context (inter- and intra-episode history) are necessary for shaping to occur. For generalizability, we conducted experiments with Llama-3.2-1B-Instruct (Appendix A.9), demonstrating consistent shaping behavior across architectures (3.96/0.10 rewards matching Gemma's 3.96/0.10).

The reviewer did not respond before the rebuttal period was truncated.

**Reviewer Dj8k**

Reviewer Dj8k was interested in generalizability to more complex scenarios. We provided a discussion addressing this, and the reviewer confirmed that they were planning to maintain their positive score (8).

References:
- Khan et al., 2024. Scaling opponent shaping to high dimensional games. In *Proceedings of the 23rd International Conference on Autonomous Agents and Multi-Agent Systems* (AAMAS’24).
- Lu et al., 2022. Model-free opponent shaping. In *Proceedings of the 39th International Conference on Machine Learning* (ICML’22).
- Meulemans et al., 2024. Multi-agent cooperation through learning-aware policy gradients. arXiv preprint arXiv:2410.18636.

---

### Meta-Review · Area_Chair_6ksb · 2026-01-08

**Summary:**

The paper studies opponent shaping in the context of LLM agents. The agents are tested in game-theoretic environments (iterated competitive zero-sum and cooperative general-sum games) where they are learning using PPO with QLoRA. The overall sentiment of the reviews was quite positive, and reviewers recognized multiple strengths of the paper including novelty and a thorough experimental analysis.

The reviewers raised concerns about some elements of presentation clarity, generalizability of the results, and potential lack of novelty. The authors addressed all the concerns raised by the reviewers, by both answering their questions and incorporating additional experiments and ablation studies into the paper (e.g., adding experiments with another base LLM). As a result, one of the reviewers withdrew their concern the about lack of novelty and change their score from 2 to 8 (according to the authors' statement, since score updates were reverted); concerns of other reviewers were addressed as well, even though they did not respond to the rebuttal.

My assessment is that this paper is a strong contribution. I share the positive sentiment of the reviews and recommend accepting it.

**Reviewer Concerns:**

- The authors accurately summarized all the raised concerns and how they addressed them:
https://openreview.net/forum?id=yJoHTqUNry&noteId=CeCPn1YEod.
- The only remaining concerns are around generalizability of the results to more complex games beyond 2x2 matrix games, or games that include natural language communication. Addressing this concern would substantially change the scope of the paper, and I find it reasonable to consider this concern as a suggestion for follow up work.

**Reviewer Scores:**

- ZruM: original score of 2 which the reviewer indicated that they will increase after the rebuttal; the authors stated that the reviewer increased the score to 8, which is very plausible given the very positive response to the rebuttal.
- X6KR: original score of 4; no response from the reviewer to the rebuttal, but all their concerns were fully addressed. My best assessment would be that the reviewer would increase the score by 1-2 points.
- ypQM: original score of 4; no response from the reviewer to the rebuttal, but all their concerns were fully addressed. My best assessment would be that the reviewer would increase the score by 1-2 points.
- Dj8k: original score of 8 which they indicated will maintain.

---

### Decision · Program_Chairs · 2026-01-26

Accept (Poster)